# Advancements in Binary Solvent-Assisted Hydrogel Composites for Wearable Sensing Applications

**DOI:** 10.3390/ma17225535

**Published:** 2024-11-13

**Authors:** Garam Choi, Fathilah Ali, Kyoungho Kim, Minsoo P. Kim

**Affiliations:** 1Department of Chemical Engineering, Sunchon National University, Suncheon 57922, Republic of Korea; chlrkfka78@gmail.com; 2Department of Chemical Engineering and Sustainability, Kulliyyah of Engineering, International Islamic University Malaysia, Kuala Lumpur 50728, Malaysia; fathilah@iium.edu.my; 3Department of Chemical Engineering, Dong-Eui Institute of Technology (DIT), Busan 47230, Republic of Korea; kkim@dit.ac.kr

**Keywords:** binary solvent, hydrogel composite, wearable sensing

## Abstract

The advancement of wearable sensing technologies has been pivotal in revolutionizing healthcare, environmental monitoring, and personal fitness. Among the diverse materials employed in these technologies, multifunctional hydrogel composites have emerged as critical components due to their unique properties, including high water content, flexibility, and biocompatibility. This review provides a comprehensive overview of the state-of-the-art in binary solvent-assisted hydrogel composites for wearable sensing applications. It begins by defining hydrogel composites and their essential attributes for wearable sensors, specifically focusing on binary solvent-assisted methods that enhance their performance and functionality. The review then delves into the applications of these composites in health monitoring, environmental detection, and sports and fitness, highlighting their role in advancing wearable technologies. Despite their promising features, there are significant challenges related to durability, sensitivity, and integration that need to be addressed to fully exploit these materials in wearable devices. This review discusses these challenges and presents potential solutions, including the development of new materials, improvement in fabrication processes, and strategies for achieving multifunctionality and sustainable design. Looking forward, the paper outlines future directions for research in this field, emphasizing the need for innovative materials and technologies that can lead to more effective, reliable, and eco-friendly wearable sensors. This review aims to inspire further research and development in the field of wearable sensing, paving the way for new applications and advancements in healthcare, environmental monitoring, and personal fitness technologies.

## 1. Introduction

Wearable sensor technology has emerged as a transformative force in the realms of healthcare, environmental monitoring, and personal fitness, offering unprecedented capabilities for continuous, real-time physiological parameter monitoring [1,2,3,4,5]. The integration of these sensors into artificial intelligence systems and healthcare devices has ushered in a new era of human–machine interaction, enabling the translation of various external stimuli—such as mechanical signals from bodily movements, temperature fluctuations, and humidity levels—into quantifiable electrical signals [6,7,8,9,10].

The exponential advancement in this interdisciplinary field has catalyzed numerous systematic reviews examining the multifaceted aspects of wearable technologies. Liu et al. provided comprehensive analyses of piezoresistive and capacitive sensing mechanisms in flexible electronics, establishing correlations between material architectures and transduction efficiency [11]. The integration of multiplexed electrochemical sensing arrays for real-time biomolecular detection has been systematically evaluated by Gao et al., emphasizing the significance of multimodal detection platforms [12]. Trung and Lee’s seminal work elucidated the structure–property relationships in stretchable electronics, particularly focusing on the mechanotransduction mechanisms and interfacial engineering [13]. Sun et al. provided detailed insights into the hierarchical assembly of functional materials for biointegrated electronics [14], while Rogers et al. comprehensively reviewed the emerging paradigms in conformable bioelectronics [15]. Moreover, Wang et al. detailed the development of flexible sensing electronics for health-monitoring applications [16]. However, these scholarly works, despite their significant contributions, have not systematically addressed the physicochemical interactions and synergistic effects in binary solvent systems—critical parameters that fundamentally influence material stability and performance metrics. This knowledge gap becomes particularly significant as binary solvent-assisted hydrogel composites emerge as a promising platform for addressing fundamental challenges in environmental stability, mechanical robustness, and operational consistency under extreme thermomechanical conditions.

The advancement of wearable sensor technologies, while promising, faces significant challenges in maintaining reliable performance under real-world operating conditions. A primary concern is the environmental stability of sensing materials, particularly when exposed to the diverse conditions encountered in daily use [17,18]. Traditional sensing materials often exhibit performance degradation under varying temperatures and humidity levels, limiting their practical applications in continuous monitoring scenarios [19,20,21]. This challenge is particularly pronounced in hydrogel-based sensors, which, despite their excellent biocompatibility and flexibility, are susceptible to dehydration, freezing, and mechanical failure under ambient conditions [22,23]. Specifically, the requirements for real-time monitoring in healthcare and environmental applications have highlighted several critical challenges in sensor development. Environmental resilience remains a fundamental concern, as sensors must maintain consistent performance across diverse conditions, from sub-zero temperatures to high-humidity environments [24,25]. The need for long-term stability presents another significant challenge, as continuous monitoring applications demand sustained functionality without signal drift or material degradation [26,27]. These limitations have particularly impacted the development of flexible strain sensors, which require both mechanical durability and consistent performance under various environmental conditions.

Flexible strain sensors, in particular, have garnered significant attention due to their versatility and applicability in diverse fields, including electronic skin, wearable health-monitoring systems, soft robotics, and implantable medical devices [2,3,4,5,28,29,30,31,32,33,34]. The evolution of these sensors has been characterized by the incorporation of advanced materials and innovative design approaches, with hydrogel composites emerging as particularly promising candidates [28,35,36,37,38].

Hydrogels, defined by their water-rich, three-dimensional network structures, offer an ideal platform for flexible sensors due to their high flexibility, stretchability, and excellent biocompatibility. These properties facilitate seamless integration with human tissues, thereby creating novel interfaces between biological systems and technology [17]. However, conventional hydrogels face significant challenges related to their environmental stability. The hydrophilic network structure of these materials tends to become rigid and brittle when exposed to extreme temperatures, primarily due to water evaporation or freezing [18,19,20]. Moreover, prolonged exposure to dry environments leads to a degradation of elasticity and conductivity as moisture evaporates from the matrix [17,21]. These limitations severely impair the conductive, flexible, and mechanical properties of hydrogels, thus restricting their durability and application potential in complex environments.

To address these issues, recent research has focused on developing hydrogels with enhanced environmental resilience [22,23]. A critical aspect of this development involves the prevention or deceleration of water freezing and evaporation within the hydrogel matrix [24,25,26,27,39,40,41]. The design of such hydrogels is essential for expanding their applicability in flexible sensors and biomedical fields. Despite the significant impact of solvents on hydrogel performance, this parameter has not received sufficient attention in the research community. Consequently, there is a pressing need to design dual-solvent conductive hydrogel systems that incorporate inherent antifreeze properties prior to crosslinking, capable of retaining moisture for extended periods and offering resistance to extreme temperatures.

Solvents such as glycerol, ethylene glycol, dimethyl sulfoxide (DMSO), butanediol (BDO), and ionic liquids have demonstrated significant potential in reducing the freezing point of hydrogels, thereby enhancing their low-temperature stability [24,25,26,27,39,42,43]. These solvents facilitate the production of hydrogels with antifreeze characteristics, which are particularly advantageous for wearable device applications (Figure 1). The strong hydrogen bonding between the solvent and water competes with water’s own hydrogen bonding [44], enhancing the hydrogel’s water retention capacity and effectively inhibiting water freezing. This contributes to the hydrogel’s elasticity and mechanical robustness. Moreover, the incorporation of a binary solvent system introduces additional non-covalent interactions within the polymer network [45]. These interactions function as sacrificial bonds, effectively dispersing external energy when the hydrogel undergoes deformation. Consequently, this mechanism contributes to enhancing the overall strength and durability of the hydrogel material.

This review aims to provide a comprehensive survey of the latest trends, challenges, and future research directions in the realm of wearable sensing technologies that utilize hydrogel composites. By comparing and analyzing the mechanical properties, antifreeze characteristics, water retention capabilities, and conductivity of traditional single-solvent and dual-solvent hydrogels, this review seeks to offer a nuanced understanding of the current state and potential advancements in this field. Additionally, the review explores potential applications based on the unique properties of environmentally resilient hydrogels, including wearable sensors, bioelectrodes, soft robotics, and wound dressings, offering insights into future research directions and development prospects.

## 2. Binary Solvent-Assisted Hydrogel Composites

In the rapidly evolving field of materials science, hydrogel composites have emerged as a promising class of materials with diverse applications, particularly in the realm of wearable sensing technologies. These three-dimensional polymeric networks, capable of retaining substantial amounts of water while maintaining their structural integrity, have garnered significant attention due to their unique properties and potential for customization [46,47,48].

The incorporation of binary solvents and functional additives crucially modifies hydrogel properties through several key mechanisms that address fundamental limitations of conventional single-solvent hydrogels. These modifications are particularly important for wearable sensing applications, where environmental stability and consistent performance are essential. First, binary solvent systems fundamentally alter the freezing behavior and mechanical properties of hydrogels. For example, in glycerol/water systems [25], glycerol molecules form extensive hydrogen bonding networks that effectively prevent water crystallization while maintaining polymer chain mobility. This mechanism enables remarkable mechanical properties (fracture toughness ~2300 J/m^2^) and stable conductivity (8.2 S/m) even at −20 °C, far exceeding the capabilities of traditional water-only hydrogels that become rigid and non-conductive at sub-zero temperatures. Second, binary solvents enhance hydrogel stability across diverse environmental conditions. DMSO/water combinations [49] demonstrate exceptional resistance to both freezing and dehydration, maintaining functionality from −50 °C to room temperature. This remarkable stability stems from DMSO’s unique ability to interact simultaneously with water molecules and polymer chains, creating a more robust network structure. The result is a hydrogel that exhibits both high mechanical performance (tensile strength: 0.89 MPa, elongation: 696%) and consistent ionic conductivity across extreme temperature ranges. Third, the addition of functional materials in binary solvent systems creates synergistic effects that further enhance performance. For instance, the incorporation of conductive nanoparticles in PVA/PANi hydrogels [27] leads to significantly improved electrical properties (conductivity: 0.415 S/m at room temperature, 0.32 S/m at −20 °C) while maintaining excellent mechanical characteristics (fracture strain: 472%). The binary solvent environment enables a better dispersion and integration of these functional additives, resulting in more uniform and stable composite materials. The preference for binary solvent systems is further justified by their ability to enable multiple functionalities simultaneously. For example, ethylene glycol/water (GW)-based hydrogels [50] demonstrate remarkable strain-sensing capabilities across an unprecedented temperature range (−55.0 °C to 44.6 °C) while maintaining structural integrity and consistent electrical performance. This multifunctional capability is particularly crucial for wearable sensing applications, where devices must perform reliably under varying environmental conditions. These modifications address critical limitations of conventional hydrogels, such as poor mechanical strength, limited conductivity, and susceptibility to environmental factors like freezing and dehydration, which have historically restricted their practical applications [22,23,51].

Building on these demonstrated advantages, the development of binary solvent-assisted hydrogel composites has been systematically optimized [25,26,27,39] (Table 1), establishing them as ideal candidates for advanced wearable sensing applications (Figure 1). The optimization process focuses on three key components: First, the polymer matrix, which forms the structural backbone of the hydrogel, typically employs materials such as polyvinyl alcohol (PVA) or polyacrylamide (PAM), chosen specifically for their influence on mechanical properties and biocompatibility. Second, while water serves as the primary solvent providing characteristic hydrophilicity, the careful selection of secondary organic solvents like glycerol, ethylene glycol, or dimethyl sulfoxide (DMSO) enables precise tuning of crucial properties. These organic solvents often act as plasticizers, enhancing flexibility and preventing crystallization at low temperatures. Additionally, functional additives like nanoparticles, conductive polymers, or ionic species can be incorporated to impart specific properties such as electrical conductivity or sensing capabilities [52].

The fabrication of binary solvent-assisted hydrogel composites involves several advanced techniques (Figure 2), each offering unique advantages in terms of scalability, control over material properties, and compatibility with various hydrogel compositions. One of the most straightforward methods is the one-pot polymerization technique. This approach involves the simultaneous polymerization and crosslinking of monomers in the presence of both solvents. For instance, Yu et al. employed this technique to create a highly stretchable and conductive hydrogel using glycerol/water as the binary solvent system (Figure 2a) [53]. The resulting hydrogel exhibited remarkable mechanical properties and electrical conductivity, showcasing the potential of this fabrication method.

Another popular fabrication technique is UV-initiated radical polymerization. This approach utilizes ultraviolet light to initiate the polymerization process, allowing for precise control over the reaction kinetics. Jung et al. demonstrated the efficacy of this method in producing temperature-tolerant hydrogels for supercapacitor applications (Figure 2b) [54]. The UV-initiated process enabled the researchers to fine-tune the crosslinking density and incorporate functional additives effectively, resulting in hydrogels with excellent thermal stability and electrical properties.

**Figure 1 materials-17-05535-f001:**
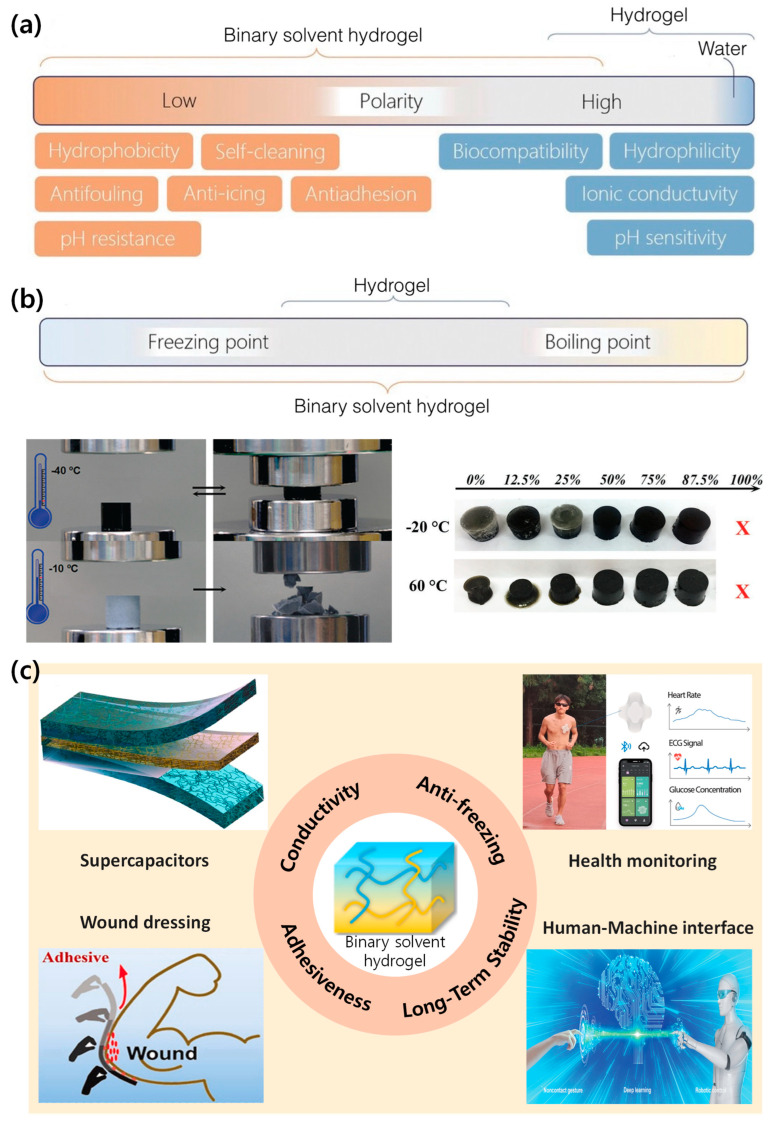
Properties and applications of binary solvent hydrogels. (**a**) Characteristics of binary solvent hydrogels compared to water-based hydrogels. (**b**) Temperature property of binary solvent hydrogels, demonstrating stability at extreme temperatures. (Reproduced with permission from Ref. [50]. Copyright 2017 John Wiley and Sons). (**c**) Key properties and diverse applications of binary solvent hydrogels, including supercapacitors, health monitoring, and wound dressing. (Reproduced with permission from Ref. [55]. Copyright 2022 Elsevier. Reproduced with permission from Ref. [56]. Copyright 2024 Elsevier. Reproduced with permission from Ref. [57]. Copyright 2022 Elsevier. Reproduced with permission from Ref. [58]. Copyright 2022 John Wiley and Sons).

**Table 1 materials-17-05535-t001:** Characteristics of binary solvent-assisted hydrogel composites.

Materials	Binary Solvent	Preparation Method	Mechanical Properties	Antifreezing Temperature	Conductivity	Ref.
^1^ PDA-CNT	Glycerol/water	UV-initiated copolymerization	Maximum tensile strain ~700% Fracture toughness ~2300 J/m^2^ Adhesive strength ~57 ± 5.2 kPa	Maintained performance at −20 °C	8.2 S/m	[25]
Guar gum/Borax	Glycerol/water	-	Adhesion strength: 2.5 kPa Swelling ratio: ~25% after 10 h	Maintained flexibility at −16 °C	-	[26]
^2^ PVA/PANi	Glycerol/water	cyclic freezing and thawing	Fracture stress/strain/energy: 280 kPa/472%/1.88 MJ/m^3^ Young’s modulus: 30.2–34.5 kPa	Maintained flexibility and conductivity at −20 °C Freezing point lowered by ~42 °C compared to single-solvent gel	Maximum conductivity: 0.415 S/m 0.32 S/m at −20 °C	[27]
^3^ OEGMA/MEO_2_MA/Clay	Glycerol/water	in situ polymerization	Elongation: Up to 600% Tensile stress at −10 °C: 15–133 kPa (depending on glycerol content) Fracture energy at −10 °C: 1139–2137 J/m^2^ (depending on glycerol content)	Maintained flexibility at −20 °C (for 30 wt% glycerol content) Freezing point: −43.7 °C (cooling), −21.8 °C (heating) for 30 wt% glycerol content	Maximum conductivity: 3.32 × 10^−4^ S/cm (0 wt% glycerol) Conductivity at −20 °C: 2.4 × 10^−4^ S/cm (30 wt% glycerol)	[39]
PVA/EG	Ethylene glycol/water	one-pot polymerization	Tensile strength: 1.1 MPa Elongation at break: 442.3% Compressive strength: 2.4 MPa at 80% strain	Maintained flexibility at −20 °C	0.173 S/m at 25 °C 0.156 S/m at −20 °C	[42]
(PVA)/CNC-^4^ PDDA/PA	Glycerol/water	freeze-thaw cycle	Tensile stress: 1.0 MPa at −30 °C Strain: 513.8% at −30 °C Self-healing efficiency: 85.0% at −30 °C	Maintained flexibility at −30 °C Freezing point: −38 °C (for 25 wt% phytic acid content)	0.6 S/m at −30 °C	[43]
^5^ PEDOT:PSS	Ethylene glycol/water	one simple freezing step	Tensile strength: 52 kPa–2.1 MPa Elongation at break: 200–900% Young’s modulus: 14–330 kPa	Maintained flexibility and strain sensitivity from −55.0 °C to 44.6 °C (for optimal 66.6 wt% EG composition) Functional down to −40 °C	Maintained stable electrical conductivity at −40 °C	[50]
PVA/^6^ SNF/CN	Ethylene glycol/water	one-pot polymerization	Tensile strength: 1.39 MPa Strain: 585.8% Toughness: Increased by 7.7 times compared to pure PVA hydrogel	Maintained flexibility at −18 °C No freezing peak observed from 25 °C to −70 °C in DSC measurement	2.85 mS/cm at RT 2.07 mS/cm at −18 °C	[59]
PVA/^7^ CNF	DMSO/water	one-pot method	Tensile strength: 0.89 MPa Elongation at break: 696% Young’s modulus: 0.29 MPa Toughness: 3.54 MJ/m^3^ Compressive stress at 90% strain: 11.01 Mpa	Maintained flexibility at −50 °C No freezing peak observed between 30 °C and −80 °C in DSC measurement	Maintained ionic conductivity at −50 °C	[49]
PVA/^8^ LN	DMSO/water	freezing-thawing method	Tensile strength: 1.28 MPa Elongation at break: Not explicitly stated Toughness: 1.9 MJ/m^3^ Young’s modulus: 0.52 MPa Compressive strength: 18 MPa at 90% strain	Maintained flexibility at −80 °C No freezing peak observed between −90 °C and 100 °C in DSC measurement	0.162–0.175 mS/cm (between −80 °C and 20 °C)	[60]
^9^ CMC/MEA	DMSO/water	radical polymerization	Maximum tensile strain: 864.74% Maximum tensile stress: 83.57 kPa Maximum compressive stress: 141.89 kPa	No ice crystallization observed from −90 °C to 20 °C for Gel 5:5 (5:5 ratio of water/DMSO) Freezing point decreased to −17 °C for Gel 7:3 (7:3 ratio of water/DMSO)	-	[61]
PVA	Ethanolic ferric chloride	freeze-casting-assisted solution substitution (FASS)	Young’s modulus: 1.69 MPa Tensile strength: 5.75–6.77 MPa Maximum tensile strain: 1400–1710% Toughness: 55.6–59.4 MJ/m^3^ Fracture energy: 661.54 kJ/m^2^	Freezing tolerance down to −30 °C, maintaining some conductivity and flexibility at this temperature	1.3–6.5 S/m Conductivity could be tuned by adjusting the ferric chloride concentration from 1 to 8 wt%	[62]
F127	[EMIm]Cl/water	radical polymerization under UV light irradiation	Tensile strength: 139 kPa Fracture strain: 1835% Cyclic stability: Maintained stable strength over 1000 cycles at 100% tensile strain	No crystallization peak observed when [EMIm]Cl content exceeds 30 wt% Maintained flexibility of over 1300% strain at −20 °C	0.28 S/m at 25 °C 0.092 S/m at −20 °C	[63]
^10^ PAA-co-DMAPS	[EMIM][OAc]/water	one-step copolymerization	Tensile strength: ~1 MPa Strain: ~600% Young’s modulus: 77–142 kPa Toughness: 1883.63 kJ/m^3^	Maintained flexibility and transparency down to −40 °C No crystallization peak observed in DSC for sample with 35 wt% ionic liquid, indicating freezing tolerance below −80 °C	12.40 mS/cm at RT 0.87 mS/cm at −30 °C	[64]
PVA	[EMIm]Ac/water	two-step copolymerization	Breaking strain: >350% at 25 °C, ~250% at −50 °C Elastic modulus: 174 ± 21 kPa at 25 °C Compressive modulus: 112 ± 10 kPa at 25 °C Toughness: 600–800 kJ/m^3^	Freezing point: −65 °C (for PVA/EMImAc/H_2_O hydrogel) Maintained flexibility and conductivity down to −50 °C	2.98 S/m at 25 °C 0.22 S/m at −50 °C	[65]

^1^ PDA: Polydopamine; ^2^ PVA/PANi: Polyvinyl alcohol/polyaniline; ^3^ OEGMA/MEO_2_MA: oligo(ethylene glycol) methacrylate/(2-methoxyethoxy) ethyl methacrylate; ^4^ PDDA/PA: Poly(diallyldimethylammonium chloride)/phytic acid; ^5^ PEDOT:PSS: Poly (3,4-ethylenedioxythiophene)/poly(styrene sulfonate); ^6^ SNF/CN: Silk nanofibers/graphitic carbon nitride nanosheets; ^7^ CNF: Cellulose nanofiber; ^8^ LN: Lignin nanoparticle; ^9^ CMC/MEA: Carboxymethyl cellulose sodium salt/2-methoxyethyl acrylate; ^10^ PAA-co-DMAPS: Poly(acrylic acid)-co-zwitterionic monomer 3-dimethyl(methacryloyloxyethyl)ammonium propane sulfonate.

**Figure 2 materials-17-05535-f002:**
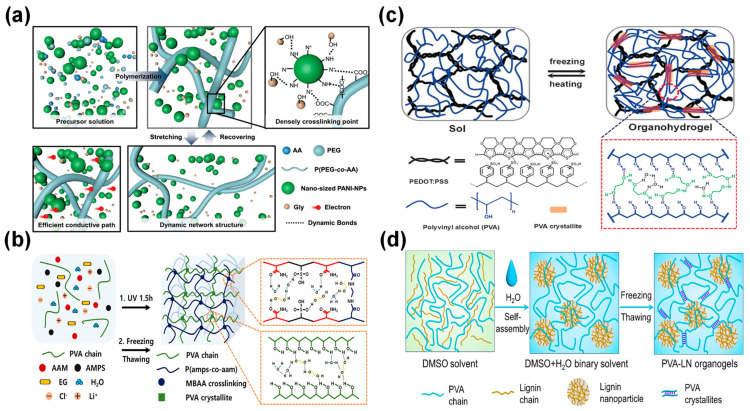
Schematic illustrations of various binary solvent-assisted hydrogel composites and their preparation methods. (**a**) Dynamic crosslinked networks with nano-sized PANI-NPs. (Reproduced with permission from Ref. [53]. Copyright 2022 John Wiley and Sons). (**b**) Fabrication of organohydrogel showing chemical and physical crosslinking. (Reproduced with permission from Ref. [54]. Copyright 2022 Elsevier). (**c**) Preparation of PEDOT:PAA-PVA organohydrogel using H_2_O/EG binary solvent. (Reproduced with permission from Ref. [50]. Copyright 2017 John Wiley and Sons). (**d**) Preparation process of antifreezing conductive PVA-LN organogels via solvent-induced self-assembly and freeze–thaw method. (Reproduced with permission from Ref. [60]. Copyright 2022 Elsevier).

For hydrogels requiring enhanced mechanical properties, the freeze–thaw cycling method has proven to be particularly effective. This physical crosslinking technique involves repeated freezing and thawing of the hydrogel precursor solution, promoting the formation of crystalline regions that act as physical crosslinks. Rong et al. utilized this technique to develop antifreezing conductive hydrogels with excellent mechanical properties (Figure 2c) [50]. The freeze–thaw process resulted in a unique microstructure that contributed to the hydrogel’s improved strength and flexibility.

In cases where nanostructured hydrogels are desired, the self-assembly and sol–gel transition method offers a powerful bottom-up approach. This technique leverages the self-assembly properties of certain molecules to create intricate nanostructures within the hydrogel matrix. Feng et al. employed this method to fabricate lignin nanoparticle-reinforced hydrogels with exceptional freeze resistance (Figure 2d) [60]. The self-assembly process allowed for the precise control of the hydrogel’s nanostructure, resulting in enhanced mechanical and thermal properties.

Binary solvent systems fundamentally transform hydrogel composite properties through synergistic molecular interactions, enabling unprecedented performance in wearable sensing applications [24,25]. These systems specifically enhance mechanical stability, electrical conductivity, and environmental resilience through controlled solvent–polymer interactions [26,27,39]. Recent studies have demonstrated remarkable temperature stability from −50 °C to 60 °C, with conductivity retention reaching 95% at −20 °C and strain sensitivity maintaining a gauge factor of ~2 even at extreme temperatures [27,39]. Furthermore, these advanced composites exhibit self-healing efficiency exceeding 90% across their entire operational range, representing a significant advancement over conventional single-solvent systems [43]. One of the most widely studied binary solvent systems is the glycerol/water combination (Figure 3a). Hydrogels based on this system exhibit high conductivity, excellent stretchability, and remarkable toughness. For example, Jiang et al. reported a mussel-inspired hydrogel with a conductivity of 8.2 S/m, stretchability of 700%, and toughness of 2300 J/m^2^ (Figure 3b) [25]. The glycerol component plays a crucial role in preventing water crystallization at low temperatures, enabling these hydrogels to maintain their performance across a wide temperature range, typically from −20 °C to 60 °C. This exceptional stability in extreme temperatures, coupled with superior mechanical properties, makes glycerol/water-based hydrogels particularly suitable for outdoor wearable sensing applications.

Another promising binary solvent system is the ethylene glycol/water combination (Figure 3c). Hydrogels based on this system possess high strength, stretchability, and mechanical stability. Bao et al. reported a PVA/SNF/CN hydrogel with a tensile strength of 1.39 MPa and elongation at break of 585.8% [59]. The ethylene glycol component not only contributes to the hydrogel’s antifreezing properties but also helps maintain its conductive properties at low temperatures. These hydrogels exhibit rapid response times as strain sensors, making them ideal for real-time motion detection applications (Figure 3d–f). Li et al. demonstrated a PVA/EG hydrogel with a high gauge factor of 0.725, showcasing its potential for highly sensitive flexible sensing applications (Figure 4a,b) [42].

For applications requiring extreme freeze resistance and dehydration prevention, the DMSO/water binary solvent system has shown remarkable results. Li et al. reported an ionic conductive hydrogel with stretchability up to 696% and toughness of 3.54 MJ/m^3^ (Figure 4c–f) [49]. The DMSO component significantly lowers the freezing point of the hydrogel, allowing it to maintain flexibility and functionality at temperatures as low as −50 °C. This wide operating temperature range (−50 °C to 50 °C) makes DMSO/water-based hydrogels particularly suitable for applications in harsh environmental conditions.

In recent years, ionic liquid/water systems have gained attention due to their unique properties. Liu et al. reported an ionic hydrogel with a conductivity of 0.28 S/m at 25 °C and transparency of 94.26% (Figure 5a–c) [63]. The ionic liquid component not only contributes to the hydrogel’s high ionic conductivity but also imparts excellent freeze resistance and long-term dehydration resistance. Zhang et al. demonstrated an ionic hydrogel with a strain-sensing range of 0–300% and a linear gauge factor of 1.93 (Figure 5d–f) [64], highlighting the potential of these materials for precise strain-sensing applications.

Despite these promising characteristics, the practical implementation of ionic liquid-based hydrogels necessitates a meticulous consideration of their chronological stability mechanisms. The exceptional performance metrics observed in these systems are fundamentally governed by the intermolecular interactions and electrochemical dynamics between ionic moieties and the polymeric matrix. However, these physicochemical interactions undergo temporal evolution through multiple degradation pathways. Welton’s fundamental investigations elucidated that ionic liquids containing nucleophilic anions experience progressive structural reorganization, particularly affecting the quaternary ammonium cation configurations crucial for maintaining ionic conductivity [66]. MacFarlane et al. further established that these conformational alterations substantially influence the ionic transport kinetics and viscoelastic properties of the hydrogel network [67]. Environmental parameters introduce multifaceted complexity to the stability paradigm. Armand and colleagues demonstrated that atmospheric moisture and oxygen exposure initiate cascade hydrolysis mechanisms, predominantly in hygroscopic ionic liquid systems [68]. These mechanochemical reactions, as systematically analyzed by Swatloski et al., induce a progressive deterioration of both electrochemical conductivity and rheological properties [69]. Moreover, aromatic cation-based architectures prevalent in hydrogel systems exhibit pronounced susceptibility to photochemical decomposition, necessitating strategic considerations in molecular design implementation. Recent innovations in supramolecular architecture have initiated the resolution of these stability challenges. Plechkova and Seddon demonstrated that the strategic modification of ionic liquid conformations, specifically through the incorporation of sterically hindered cationic species, substantially enhances their environmental resilience [70]. Wasserscheid and Welton’s comprehensive analysis established that the precise selection of ionic pairs, coupled with targeted stabilizing additives, effectively suppresses these degradation mechanisms while preserving desired functionalities [71]. These advanced molecular engineering strategies have established novel paradigms for developing thermodynamically stable ionic liquid-based hydrogels, although achieving economic viability in large-scale implementation remains a significant consideration.

While each binary solvent system offers unique advantages, they also come with specific challenges that researchers must address. For instance, the potential toxicity of ethylene glycol may limit its use in certain biomedical applications. Similarly, the ability of DMSO to penetrate biological membranes raises concerns for some biomedical uses. The high cost of ionic liquids and potential long-term stability issues are factors that need consideration when developing ionic liquid-based hydrogels. Despite these challenges, the field of binary solvent-assisted hydrogel composites continues to advance rapidly. Researchers are exploring new solvent combinations, novel fabrication techniques, and innovative functional additives to further enhance the properties of these materials. The integration of nanotechnology, such as the incorporation of graphene or carbon nanotubes (CNTs), is opening up new possibilities for creating hydrogels with unprecedented electrical and mechanical properties [72,73,74].

Looking to the future, the development of multifunctional binary solvent-assisted hydrogel composites holds great promise. These advanced materials could simultaneously sense multiple stimuli, such as strain, temperature, and biochemical markers, paving the way for next-generation wearable health-monitoring devices. Additionally, the combination of sensing capabilities with drug delivery functions could lead to smart wound dressings that not only monitor the healing process but also deliver therapeutic agents as needed [75]. Binary solvent-assisted hydrogel composites represent a significant advancement in materials science, offering enhanced properties and expanded functionalities for wearable sensing applications. By carefully selecting the solvent system, fabrication technique, and functional additives, researchers can tailor these materials to meet the diverse requirements of next-generation flexible electronics and biomedical devices. As this field continues to evolve, we can expect to see increasingly sophisticated hydrogel-based sensors that push the boundaries of what is possible in wearable technology, potentially revolutionizing areas such as healthcare, sports science, and human–machine interfaces.

## 3. Wearable Sensing Applications

The advent of binary solvent-assisted hydrogels has opened up a new frontier in wearable sensing technologies, enabling applications that were previously challenging or impossible with conventional materials. These advanced hydrogels, with their unique properties, are revolutionizing various fields including healthcare monitoring, sports performance analyses, and human–machine interfaces. This section explores the diverse and innovative applications of these materials in wearable sensing (Table 2).

### 3.1. Continuous Health Monitoring

The integration of binary solvent-assisted hydrogels has revolutionized healthcare monitoring through unprecedented real-time capabilities. These materials achieve response times below 100 ms while simultaneously detecting multiple physiological parameters including temperature, strain, and various biochemical markers.

A significant breakthrough in this field was demonstrated by Jiang et al., who developed a glycerol/water-based hydrogel sensor for monitoring heart rate and blood oxygen levels continuously (Figure 6a,b) [25]. This sensor exhibited remarkable performance across a wide temperature range (−20 °C to 60 °C), effectively overcoming the environmental limitations of conventional sensors. The glycerol component played a crucial role in preventing water crystallization at low temperatures, while also enhancing the sensor’s mechanical properties and conductivity. The ability to function in extreme temperatures represents a significant advancement in wearable sensor technology. It expands the potential applications of these sensors to diverse environmental conditions, from arctic expeditions to desert explorations, where traditional sensors would fail due to temperature-induced malfunction or degradation.

Furthermore, Liu et al. made substantial progress in the realm of biochemical sensing with their ionic liquid/water hydrogel sensor designed for real-time sweat analyses (Figure 6c) [63]. This multifunctional sensor demonstrated the capability to simultaneously detect glucose, lactate, and pH levels in sweat, providing a comprehensive picture of an individual’s metabolic state. The high ionic conductivity of the hydrogel (0.28 S/m at 25 °C) ensured accurate and sensitive measurements, while its excellent transparency (94.26%) allowed for easy visual inspection, potentially enabling integration with optical sensing modalities. The development of such multimodal sensors represents a significant step towards personalized health monitoring. By providing real-time data on multiple physiological parameters, these sensors could enable the early detection of health issues and facilitate timely interventions. Moreover, the continuous nature of the monitoring allows for the detection of subtle changes over time, which might be missed by periodic measurements.

However, challenges remain in the widespread adoption of these sensors. Issues such as long-term biocompatibility, sensor drift, and the need for robust data interpretation algorithms need to be addressed. Additionally, the integration of these sensors with existing healthcare systems and the development of user-friendly interfaces for data presentation are areas that require further research and development.

### 3.2. Advanced Motion Capture and Biomechanics Analysis

The field of motion capture and biomechanics analyses has been revolutionized by the advent of binary solvent-assisted hydrogels, offering unprecedented levels of comfort, accuracy, and versatility in wearable sensing applications. These advanced materials have enabled the development of sensors that can capture complex human movements with high fidelity, providing valuable insights for sports science, rehabilitation medicine, and human–computer interaction.

A notable advancement in this area was achieved by Hu et al., who developed a PVA/glycerol/polyaniline (PGA) hydrogel sensor (Figure 7a–e) [27]. This sensor demonstrated exceptional sensitivity with a gauge factor of 2.14 and a rapid response time of 230 ms in detecting human motion. The high sensitivity allows for the detection of subtle movements, while the quick response time enables real-time tracking of rapid motions. These properties make the sensor particularly suitable for applications requiring precise motion capture, such as sports performance analyses or rehabilitation monitoring. The sensor’s success in tracking complex joint movements during athletic activities represents a significant leap forward in sports science and biomechanics. By providing coaches and athletes with detailed biomechanical data, these sensors enable more precise performance optimization and injury prevention strategies. The ability to capture fine-grained motion data in real time could revolutionize training methodologies across various sports disciplines.

Building on this foundation, Yu et al. pushed the boundaries of wearable sensing technology by creating a highly stretchable hydrogel sensor with over 1000% stretchability (Figure 7f,g) [53]. This remarkable elasticity allows the sensor to be directly integrated into sports apparel without restricting the wearer’s movement. The seamless integration of sensors into clothing represents a paradigm shift in wearable technology, moving away from bulky external devices towards unobtrusive, “smart” textiles. The development of such highly stretchable sensors opens up new possibilities for comprehensive body motion analyses. By incorporating multiple sensors throughout a garment, researchers and clinicians can obtain a holistic view of body mechanics during complex movements. This could be particularly valuable in fields such as ergonomics, where understanding the interplay between different body parts during various activities is crucial.

However, several challenges remain in the widespread adoption of these advanced motion capture systems. One key issue is the development of robust algorithms for interpreting the vast amount of data generated by these sensors. Translating raw sensor data into meaningful biomechanical insights requires sophisticated signal processing and machine learning techniques. Additionally, ensuring the durability of these sensors under repeated stretching and exposure to sweat and environmental factors is crucial for their practical application.

Another area of ongoing research is the integration of these motion capture systems with other sensing modalities, such as force sensors or EMG (electromyography) sensors. Such multimodal systems could provide a more comprehensive understanding of human movement, combining kinematic data with information about muscle activation and force production. As the field progresses, we can anticipate the development of even more advanced hydrogel-based motion capture systems. Future research directions may include the incorporation of self-healing properties to enhance durability, the development of wireless and self-powered sensors for greater convenience, and the exploration of biodegradable materials for environmentally friendly disposable sensors.

### 3.3. Environmental and Personal Safety Monitoring

The versatility of binary solvent-assisted hydrogels has significantly expanded the capabilities of environmental- and personal safety-monitoring systems. These advanced materials have enabled the development of sensors that can operate reliably in diverse and challenging environments, providing critical data for safety and health applications.

A groundbreaking contribution to this field was made by Wei et al., who demonstrated a nanocomposite hydrogel sensor capable of detecting temperature changes across a wide range from −20 °C to 45 °C (Figure 8a) [39]. This broad operational temperature range is particularly significant for environmental-monitoring applications, as it allows for consistent performance in varied climatic conditions. The sub-zero temperature functionality (−20 °C to 45 °C) enables critical applications in extreme environmental monitoring, from cold-chain logistics to winter sports safety systems [39,76,77]. This expanded operational range represents a significant advancement over conventional sensors’ typical temperature limitations. The application of this technology in creating smart clothing that can alert wearers to potentially dangerous environmental conditions represents a major advancement in personal safety equipment. For workers in hazardous environments, such as firefighters or industrial workers, these sensors could provide early warnings of extreme heat or cold, potentially preventing accidents and health issues related to environmental exposure.

Moreover, the development of multifunctional hydrogel sensors with antimicrobial properties, as demonstrated by Sun et al. [76], marks a significant step forward in healthcare applications. These sensors can simultaneously monitor patient vital signs while reducing the risk of hospital-acquired infections (Figure 8b,c). This dual functionality is particularly valuable in clinical settings, where infection control is a critical concern. The ability to integrate sensing capabilities with antimicrobial properties in a single material could lead to the development of more effective and efficient medical devices. The potential applications of these sensors extend beyond personal safety and healthcare. In environmental monitoring, these sensors could be deployed in various settings to track temperature fluctuations, humidity levels, and potentially even the presence of pollutants or harmful substances. Their ability to withstand harsh conditions makes them suitable for long-term deployment in remote or inaccessible locations.

However, several challenges need to be addressed to fully realize the potential of these sensors in environmental and personal safety applications. One key issue is the need for improved specificity and selectivity in detecting different environmental parameters or contaminants. While current sensors excel in temperature sensing, expanding their capabilities to detect a wider range of environmental factors would greatly enhance their utility.

Another area for future development is the integration of these sensors with data analytics and artificial intelligence systems. By combining real-time sensor data with predictive algorithms, it may be possible to develop early warning systems for environmental hazards or health risks. This could be particularly valuable in applications such as disaster prediction or personalized health monitoring. Energy efficiency and power management also remain important considerations, especially for sensors deployed in remote locations. The development of self-powered sensors, perhaps incorporating energy-harvesting technologies, could greatly enhance the longevity and reliability of environmental-monitoring systems. As research in this field progresses, we can anticipate the development of more sophisticated and integrated environmental- and personal safety-monitoring systems. Future directions may include the exploration of biodegradable materials for environmentally friendly sensors, the development of sensors capable of detecting complex chemical compounds, and the creation of networked sensor systems for large-scale environmental monitoring.

### 3.4. Human–Machine Interfaces and Soft Robotics

Recent advances in binary solvent-assisted hydrogels have enabled precise and responsive human–machine interfaces with unprecedented performance metrics including high strain sensitivity. A significant breakthrough in this field was achieved by Zhang et al., who developed an ionic hydrogel capable of sensing both strain and temperature (Figure 9a,b) [64]. This multimodal sensing capability represents a major step forward in creating more sophisticated and responsive human–machine interfaces. The ability to simultaneously detect multiple stimuli allows for a more nuanced interpretation of user interactions, potentially leading to more intuitive and natural interfaces. The application of this technology in developing more advanced prosthetic limbs is particularly noteworthy. By providing users with a sense of touch and temperature, these hydrogel-based sensors can significantly enhance the functionality and user experience of prosthetic devices. This advancement not only improves the practical utility of prosthetics but also addresses the psychological aspect of limb loss by restoring a more natural sensory experience.

Building on this foundation, Jiang et al.’s work on bioinspired ionic skin using binary solvent hydrogels has opened up new possibilities in the field of soft robotics (Figure 7c–f) [77]. These hydrogel-based ‘skins’ enable robots to interact more safely and naturally with humans and their environment. This development is crucial for the advancement of collaborative robotics, where machines work alongside humans in various settings. The potential applications of these technologies extend far beyond prosthetics and robotics. In the field of virtual and augmented reality, hydrogel-based sensors could enable more immersive and realistic haptic feedback systems. In healthcare, they could be used to create smart bandages that monitor wound healing while providing a comfortable, flexible interface with the skin.

However, several challenges need to be addressed to fully realize the potential of these materials in human–machine interfaces and soft robotics. One key issue is the long-term stability and durability of hydrogel-based sensors, especially under repeated deformation and in varied environmental conditions. Improving the mechanical robustness of these materials without compromising their flexibility and sensitivity is an important area for future research.

Another challenge lies in the integration of these hydrogel-based sensors with existing electronic systems. Developing efficient and reliable methods for signal processing and data transmission from these sensors to other components of a robotic or prosthetic system is crucial for their practical application. The biocompatibility and potential for the biodegradation of these hydrogels also present both opportunities and challenges. While these properties make them ideal for biomedical applications, ensuring consistent performance over extended periods of use in or on the body requires further investigation. As research in this field progresses, we can anticipate the development of even more advanced hydrogel-based interfaces. Future directions may include the exploration of self-healing hydrogels for more durable interfaces, the integration of drug delivery capabilities for therapeutic applications, and the development of hydrogels with programmable mechanical properties that can adapt to different use scenarios.

### 3.5. Self-Powered Wearable Devices

The development of self-powered wearable devices represents a significant advancement in the field of wearable technology, addressing one of the primary limitations of current systems—the need for frequent battery recharging or replacement. Binary solvent-assisted hydrogels have emerged as promising materials for creating such devices, offering a unique combination of sensing capabilities and energy-harvesting or storage functions.

A notable contribution to this field was made by Jung et al., who created a hydrogel using a water/ethylene glycol binary solvent system that functions both as a strain sensor and a supercapacitor (Figure 10a,b) [54]. This dual functionality is a game-changer in the realm of wearable technology. By combining sensing and energy storage capabilities in a single material, this approach significantly reduces the complexity and size of wearable devices. The application of this technology in creating self-powered wearable health monitors demonstrates its immense potential. These devices can generate and store energy from body movements, effectively eliminating the need for external power sources or frequent battery changes. This not only enhances user convenience but also extends the potential duration of continuous monitoring, which is crucial for long-term health tracking and the early detection of medical conditions.

The ability to harvest energy from body movements opens up new possibilities for wearable technology. It allows for the development of more autonomous and sustainable wearable systems that can operate for extended periods without user intervention. This is particularly valuable in scenarios where continuous, uninterrupted monitoring is critical, such as in the management of chronic health conditions or in high-performance athletic training.

However, several challenges need to be addressed to fully realize the potential of self-powered wearable devices based on binary solvent-assisted hydrogels. One key issue is optimizing the energy-harvesting efficiency of these materials. While current systems can generate sufficient power for basic sensing and data transmission, more energy-intensive functions may require further improvements in energy conversion efficiency. Another challenge lies in balancing the energy-harvesting and sensing functions of these materials. Ensuring that the energy-harvesting mechanism does not interfere with the accuracy or sensitivity of the sensing function is crucial for the reliability of these devices. The long-term stability and durability of these self-powered systems also require further investigation. Factors such as the cyclic stability of the supercapacitor function and the mechanical durability of the hydrogel under repeated deformation need to be optimized for practical, long-term use.

To address these challenges, the recent optimization of gel-based energy-harvesting devices has focused on enhancing both energy-harvesting efficiency and device durability [79,80]. The implementation of dual-network architectures has significantly improved mechanical robustness while maintaining high transparency, as demonstrated by PVA/gelatin (PG)-based hydrogels achieving tensile strengths of 0.65 MPa and stretchability up to 350% [81]. Additionally, the integration of multiple gel units in series and parallel configurations has demonstrated scalable voltage outputs up to 100 mV under real-world conditions, addressing the need for higher power output in energy-intensive applications [81]. Environmental stability, crucial for continuous monitoring applications, has been enhanced through dual-solvent strategies utilizing glycerol/water combinations, enabling reliable operation down to −60 °C while maintaining structural and functional integrity [82]. These coordinated advances in structural design and material composition have significantly improved both the energy-harvesting capabilities and long-term stability of gel-based TENGs, though challenges remain in optimizing the balance between energy-harvesting and sensing functions.

As research in this field progresses, we can anticipate the development of more advanced and efficient self-powered wearable devices. The potential applications extend beyond simple monitoring to include therapeutic interventions, leading naturally to advances in personalized medicine and drug delivery systems.

### 3.6. Personalized Medicine and Drug Delivery

Building upon advances in self-powered sensing technologies, binary solvent-assisted hydrogels have emerged as transformative platforms in personalized medicine and drug delivery. These materials offer unique opportunities for creating smart, responsive systems that can simultaneously monitor health parameters and deliver therapeutic agents in a controlled manner, effectively bridging the gap between diagnostic and therapeutic capabilities.

A notable breakthrough in this area was achieved by Pan et al., who developed self-healing glycerol/water/borax hydrogel networks (Figure 11a,b) [26]. This work has led to the creation of smart wound dressings that represent a significant leap forward in personalized medicine. These advanced dressings can not only monitor wound healing progress but also deliver medications in response to detected infections or inflammation, personalizing treatment in real time. The ability to combine sensing and drug delivery functions in a single material system is a game-changer in healthcare. It allows for the development of closed-loop systems that can autonomously respond to changes in a patient’s condition. For instance, in the context of wound care, these smart dressings can detect early signs of infection through changes in pH or temperature, and then release appropriate antibiotics or anti-inflammatory agents directly at the site of infection.

Considering these foundational advances, binary solvent-assisted hydrogels have demonstrated remarkable versatility in addressing various therapeutic needs, from chronic disease management to acute wound healing. As Li and Mooney comprehensively reviewed, the ability to precisely control drug release kinetics while preserving therapeutic efficacy has positioned these materials at the forefront of advanced drug delivery systems [83].

The implementation of binary solvent systems has particularly revolutionized diabetes management through smart insulin delivery platforms. These systems incorporate glucose-responsive elements that enable automated insulin release with minimal lag time [84]. The binary solvent environment plays a crucial role in maintaining insulin stability and protecting against enzymatic degradation, while simultaneously providing the mechanical properties necessary for sustained delivery.

Cancer therapy has similarly benefited from these advanced hydrogel platforms, particularly through the development of pH-responsive delivery systems. Mitragotri et al. highlighted how these materials leverage the acidic tumor microenvironment to achieve targeted drug release, significantly reducing systemic toxicity while maximizing therapeutic efficacy [85]. For instance, hydrogels incorporating binary solvents have achieved sustained release of chemotherapeutic agents such as doxorubicin over periods exceeding 30 days, while maintaining drug bioactivity throughout the delivery period. The superior performance of these systems can be attributed to several key advances in delivery mechanisms enabled by binary solvent incorporation. As extensively reviewed by Tibbitt et al., these hydrogels provide enhanced protection against drug degradation, particularly for sensitive biological therapeutics [86]. The presence of binary solvents enables the formation of hierarchical pore structures and reversible crosslinking mechanisms, resulting in precisely controlled release profiles that can extend from hours to months with minimal initial burst release.

In the context of wound healing, binary solvent-assisted hydrogels have demonstrated particular promise through their ability to combine antimicrobial properties with controlled release of growth factors. Zhao et al. showed how these materials can promote tissue regeneration while preventing infection, achieving significantly accelerated healing rates in chronic wound models [87]. The synchronized delivery of multiple therapeutic agents, including platelet-derived growth factor (PDGF) and vascular endothelial growth factor (VEGF), has proven especially effective in treating complex wounds such as diabetic ulcers. Moreover, these advanced systems support simultaneous delivery of multiple therapeutic agents with independent release profiles, enabling synergistic treatment approaches. Li et al. demonstrated how carefully designed binary solvent systems can maintain separate microenvironments for different therapeutic agents, allowing for optimized delivery of each component [88]. This capability is particularly valuable in complex therapeutic scenarios requiring coordinated delivery of multiple drugs.

Looking toward the future, several promising directions emerge for the continued development of these materials. The integration of artificial intelligence for predictive drug release [89] presents opportunities for more sophisticated control over therapeutic delivery. Additionally, the development of closed-loop systems responding to multiple physiological parameters could enable truly personalized therapeutic approaches. The implementation of green chemistry principles in hydrogel design [90] will be crucial for enhancing both biodegradability and biocompatibility.

The self-healing property of these hydrogels is particularly valuable in the context of wound dressings. It ensures the integrity of the dressing over time, even as it undergoes deformation or minor damage during use. This self-repair capability can extend the effective lifetime of the dressing, reducing the frequency of dressing changes and potentially improving patient comfort and healing outcomes. Moreover, the use of binary solvent systems in these hydrogels enhances their stability and functionality across a range of environmental conditions. This is crucial for maintaining consistent drug release profiles and sensing capabilities, regardless of external temperature or humidity fluctuations. The potential applications of these smart hydrogel systems extend beyond wound care. They could be used in the management of chronic conditions such as diabetes, where continuous glucose monitoring could be combined with automated insulin delivery. In the field of pain management, these materials could enable responsive analgesic delivery based on real-time assessment of pain levels or physiological stress markers.

However, several challenges need to be addressed to fully realize the potential of these materials in personalized medicine and drug delivery. One key issue is ensuring the long-term biocompatibility and stability of these hydrogels when in contact with biological tissues. While short-term studies have shown promising results, the effects of prolonged exposure need to be thoroughly investigated.

Another challenge lies in optimizing the drug release kinetics from these hydrogels. Achieving precise control over the timing and dosage of drug release in response to specific physiological triggers is crucial for effective personalized treatment. This may require the development of more sophisticated sensing mechanisms and drug encapsulation techniques within the hydrogel matrix. The integration of these smart hydrogel systems with existing healthcare infrastructure and data management systems also presents challenges. Developing standardized protocols for data interpretation and treatment decision making based on the real-time information provided by these systems will be crucial for their widespread adoption in clinical practice. As research in this field progresses, we can anticipate the development of even more advanced and multifunctional hydrogel-based systems for personalized medicine and drug delivery. Future directions may include the incorporation of multiple drug reservoirs within a single hydrogel for combination therapies, the development of biodegradable hydrogels for temporary implants, and the integration of these systems with wireless communication technologies for remote monitoring and adjustment of treatment protocols. The potential impact of these technologies on healthcare is profound. They offer the promise of more effective, personalized treatments with reduced side effects and improved patient outcomes. As these smart hydrogel systems continue to evolve, they may fundamentally change our approach to managing a wide range of medical conditions, from acute injuries to chronic diseases.

The convergence of these advances with emerging technologies suggests a future where therapeutic interventions can be precisely tailored to individual patient needs. The integration of advanced visualization and monitoring capabilities has become increasingly important in this context, particularly for precise drug administration and therapy monitoring. This growing need for sophisticated user interfaces and real-time visualization naturally leads to the exploration of augmented reality technologies in medical applications, which we will discuss in detail in the following section.

### 3.7. Augmented Reality and Gaming

Binary solvent-assisted hydrogels are transforming the landscape of augmented reality (AR) and gaming by enabling innovative user interfaces and enhancing immersive experiences. The unique properties of these advanced materials make them well suited for creating responsive touch panels, realistic haptic feedback systems, and intuitive control schemes.

A significant breakthrough in this field is the development of ionic hydrogel-based touch panels for entertainment applications, as demonstrated by Kim et al. [91]. These highly stretchable and transparent touch panels exhibit excellent electrical and mechanical properties, with a sheet resistance of 1.6 kΩ/sq, transmittance of 98%, and the ability to withstand strains up to 1000%. The incorporation of binary solvent systems enhances the stability and performance of these hydrogel touch panels across a wide temperature range (−30 °C to 50 °C) and humidity range (20% to 90% RH). This technology opens up new possibilities for creating immersive and intuitive control interfaces in gaming, allowing users to interact with virtual environments more naturally. Moreover, binary solvent-assisted hydrogels are finding applications in the development of advanced AR and virtual reality (VR) systems. Zha et al. [78] showcased the potential of nanofibrillar poly(vinyl alcohol) ionic organohydrogels for smart contact lenses and human-interactive sensing. These hydrogel-based contact lenses can detect eye movements and blinks with high sensitivity, demonstrating a gauge factor of 1.78 for a 30% strain and a response time of less than 200 ms (Figure 12b,c). The binary solvent system imparts enhanced mechanical strength (tensile strength of 0.57 MPa) and biocompatibility to these hydrogel lenses, making them suitable for extended wear and reducing the risk of eye irritation. The integration of hydrogel-based sensors into AR/VR controllers and wearables is another promising avenue for enhancing user experiences. By leveraging the strain and pressure sensing capabilities of binary solvent hydrogels, developers can create more responsive and realistic haptic feedback systems. For instance, a hydrogel-based VR glove with a pressure sensitivity of 0.1 kPa and a response time of 50 ms could provide users with a lifelike tactile sense of virtual objects, simulating different textures and firmness levels based on the object’s properties within the virtual environment.

However, several challenges need to be addressed to fully realize the potential of binary solvent-assisted hydrogels in AR and gaming applications. Ensuring the long-term durability and stability of these hydrogel interfaces under repeated use and varying environmental conditions is crucial for their widespread adoption. Additionally, integrating these hydrogel systems with existing AR/VR hardware and software platforms may require the development of specialized connectors and APIs.

As research in this field progresses, we can anticipate the emergence of even more sophisticated hydrogel-based interfaces for immersive entertainment. Future directions may include the development of self-healing hydrogels with recovery times under 1 min, the incorporation of multiple sensing modalities within a single hydrogel device with a response time below 100 ms, and the exploration of novel binary solvent systems to further enhance the performance and functionality of these materials. The integration of binary solvent-assisted hydrogels into AR and gaming technologies holds immense potential for transforming user experiences. By enabling more natural and intuitive forms of interaction, these advanced materials are paving the way for a new era of immersive entertainment, where the boundaries between the virtual and physical worlds become increasingly blurred.

## 4. Conclusions

The advent of binary solvent-assisted hydrogel composites marks a significant paradigm shift in the fields of wearable sensing technologies. This review has elucidated the transformative impact of incorporating secondary organic solvents within hydrogel matrices, effectively addressing the inherent limitations of conventional hydrogel systems.

The advent of binary solvent-assisted hydrogel composites marks a significant paradigm shift in the fields of wearable sensing technologies. The demonstrated capabilities of these composites represent substantial advancements in material engineering and sensor technology, particularly in achieving exceptional temperature stability, high strain sensitivity, and excellent self-healing efficiency. The integration of multiple functionalities within a single material system has enabled simultaneous sensing of various physiological parameters while maintaining mechanical robustness. These achievements have directly translated into diverse practical applications, from healthcare monitoring to responsive human–machine interfaces. The development of self-powered capabilities through energy harvesting from human movement marks a crucial step toward autonomous wearable systems. Such comprehensive performance characteristics establish these materials as a foundational platform for next-generation wearable technologies, particularly in applications requiring environmental resilience and multifunctional capabilities. This synergistic interaction between material properties and functional performance has significantly expanded the operational parameters of hydrogel-based sensors, enabling their deployment in diverse and challenging environments.

The development of binary solvent-assisted hydrogel composites has led to remarkable advances in material performance and functionality. As demonstrated through various binary solvent systems (Figure 3a,b for glycerol/water, Figure 3c,d for ethylene glycol/water, Figure 4c–f for DMSO/water, and Figure 5a–c for ionic liquid/water systems), carefully engineered molecular interactions enable unprecedented achievements in multiple performance metrics. These materials demonstrate exceptional temperature stability across an operational range from −50 °C to 60 °C while maintaining conductivity and mechanical integrity, as evidenced by the comprehensive performance data summarized in Table 1. The systematic analysis of different binary solvent combinations has resulted in sensors with high strain sensitivity (gauge factors exceeding 2.0), excellent self-healing efficiency (above 90%), and rapid response times (below 100 ms) for real-time physiological monitoring.

The synergistic interaction between the primary aqueous phase and the secondary organic solvent has yielded hydrogel composites with remarkable cryoprotection and hygrostability. This enhanced environmental resilience substantially expands the operational parameters of hydrogel-based sensors, enabling their deployment in diverse and challenging environments. Concomitantly, these novel composites exhibit superior mechanical properties, characterized by exceptional elasticity, enhanced toughness, and rapid self-healing capabilities. Such attributes are paramount for the development of robust and reliable wearable sensing platforms capable of withstanding the rigors of daily use.

A hallmark of these binary solvent-assisted hydrogels is their inherent multifunctionality. By seamlessly integrating various sensing modalities with additional functionalities such as electrical conductivity and antimicrobial properties, these materials serve as versatile platforms for comprehensive physiological and environmental monitoring. This multifunctionality is further augmented by advanced fabrication techniques, including UV-initiated radical polymerization and cryogenic cycling, which afford precise control over the hydrogels’ micro- and nanostructure. The unique physicochemical properties of these hydrogel composites have facilitated their application across a broad spectrum of wearable technologies. From continuous health monitoring and biomechanical motion capture to environmental sensing and human–machine interfaces, these materials are at the forefront of innovation in wearable devices. Moreover, their potential for energy storage and harvesting presents intriguing possibilities for self-powered wearable systems, addressing a critical challenge in long-term continuous monitoring. Especially in the biomedical fields, the biocompatibility and functionality of these hydrogels have opened new avenues in theranostics and personalized medicine. Applications such as smart wound dressings and controlled drug delivery systems exemplify how these materials transcend simple sensing to actively contribute to therapeutic interventions.

These fundamental material advances have directly translated into practical applications across diverse fields. In healthcare monitoring, as shown in Figure 6a–c, these materials enable the precise detection of multiple physiological parameters simultaneously, while maintaining stable performance under varying environmental conditions. Their exceptional environmental resilience, demonstrated in Figure 8a–c, has enabled reliable sensing in extreme conditions. In human–machine interfaces, their rapid response characteristics and mechanical adaptability have enabled more intuitive and reliable interactions, as illustrated in Figure 9a–f. The development of self-powered capabilities through efficient energy harvesting, shown in Figure 10a,b, has opened new possibilities for autonomous operation in long-term monitoring applications.

While the development of binary solvent-assisted hydrogel composites represents a significant advancement, it is imperative to recognize that this field is still in its developing stages. Myriad opportunities for further optimization and exploration remain, particularly in areas such as long-term stability, sensor sensitivity, and large-scale manufacturability. Binary solvent-assisted hydrogel composites have emerged as a promising platform for next-generation wearable sensors. Their unique combination of environmental stability, mechanical robustness, and multifunctionality positions them at the vanguard of wearable technology innovation.

The convergence of these capabilities—from molecular-level design (Figure 2) to system-level integration (Figure 9, Figure 10, Figure 11 and Figure 12)—positions binary solvent-assisted hydrogel composites as a transformative platform for next-generation wearable technologies. The systematic organization of fabrication methods (Figure 2a–d) and performance characteristics (Table 1 and Table 2) provides clear guidelines for future development. Looking forward, continued advancement in fabrication techniques, material optimization, and integration strategies will further expand their practical applications. Key areas for future development include enhancing long-term stability, improving energy efficiency, and developing scalable manufacturing processes, all while maintaining the exceptional performance characteristics that make these materials unique.

## 5. Future Perspectives and Challenges

As the frontier of binary solvent-assisted hydrogel composites continues to advance, material development remains a critical focus. Current technological challenges emphasize the need for improved material stability, as present systems require enhanced durability for long-term applications. Energy efficiency in active sensing modes and manufacturing scalability present additional critical challenges, requiring evolution from laboratory-scale to industrial-scale production. Research priorities follow a strategic progression across multiple timeframes. Near-term objectives focus on enhancing fundamental material properties including self-healing efficiency, response time, and operational temperature range. Mid-term goals target the development of biodegradable platforms and AI-integrated monitoring systems, while reducing production costs. Long-term directions emphasize fully autonomous systems and seamless neural interface integration, particularly for personalized medicine applications. These development trajectories, requiring coordinated advances across materials science and manufacturing processes, will facilitate the transition from current prototypes to widespread practical implementation in next-generation wearable devices. The integration of these capabilities with emerging technologies in artificial intelligence and advanced manufacturing processes presents opportunities for revolutionary advances in wearable sensing applications. This progression toward more sophisticated and integrated systems will continue to drive innovation in healthcare, environmental monitoring, and human–technology interaction.

### 5.1. Material Advancements

The frontier of binary solvent-assisted hydrogel composites is characterized by the pursuit of advanced material design and synthesis. At the forefront of this endeavor is the exploration of novel binary solvent systems, which holds the promise of hydrogels with enhanced cryoprotection, moisture retention, and mechanical properties. This research direction is increasingly aligned with the growing emphasis on sustainability in materials science, driving the investigation of green solvents and bio-based alternatives.

Concurrent with solvent innovation is the integration of nanomaterials into hydrogel matrices. The incorporation of materials such as graphene, carbon nanotubes, or metal nanoparticles presents a significant opportunity to enhance the electrical, mechanical, and sensing properties of hydrogels. However, this integration is not without challenges, as optimizing the dispersion and interaction of these nanomaterials within the hydrogel network remains a complex task requiring innovative approaches.

The development of stimuli-responsive hydrogels represents another exciting avenue in material advancement. These sophisticated materials, capable of responding to multiple environmental cues such as pH, temperature, or light, could enable the creation of more adaptive and intelligent wearable sensors. This research direction is complemented by efforts in bioinspired design, where natural systems such as the water-retaining mechanisms of desert plants or the adhesive properties of marine organisms serve as inspiration for novel hydrogel architectures with enhanced functionality and biocompatibility.

Underpinning these advancements is the ongoing challenge of enhancing the durability and stability of hydrogel-based sensors. Researchers are striving to develop materials that can maintain their performance under various environmental conditions, including extreme temperatures, high humidity, and prolonged UV exposure. This focus on long-term reliability is crucial for the practical application of hydrogel sensors in real-world scenarios.

### 5.2. Toxicological Considerations and Green Alternatives

Building upon the fundamental material advances discussed above, addressing the toxicological implications of binary solvent systems has emerged as a critical challenge in the field. While traditional organic solvents have demonstrated excellent performance in enhancing hydrogel properties, their potential cytotoxicity presents significant barriers to widespread adoption, particularly in biomedical applications [83]. This challenge has sparked innovative research into alternative, biocompatible solvent systems that maintain crucial functionality while minimizing biological risks.

Natural deep eutectic solvents (NADESs) have emerged as particularly promising alternatives in this context. These systems, composed of natural primary metabolites including sugars, amino acids, organic acids, and choline derivatives, can achieve a comparable depression of freezing points while exhibiting significantly reduced cytotoxicity compared to conventional organic solvents [92]. For instance, choline chloride-based eutectic systems have demonstrated excellent biocompatibility while maintaining functional properties, representing a significant advancement in green solvent design [93].

Bio-derived solvents represent another significant advancement in addressing toxicity concerns. The development of bio-based alternatives has been driven by green chemistry principles, focusing on materials derived from renewable resources [94]. These alternatives often demonstrate reduced cytotoxicity and enhanced biodegradability, making them particularly suitable for biomedical applications. Furthermore, the emergence of bio-ionic liquids derived from natural sources has opened new avenues for developing environmentally benign solvent systems, offering unique combinations of properties including low toxicity and excellent stability [95]. Emerging processing strategies such as supercritical CO_2_ treatment and mechanochemical approaches have also gained attention as methods to minimize exposure to potentially toxic solvents during hydrogel synthesis [96]. These techniques can significantly reduce the environmental impact of hydrogel production while maintaining desired material properties. The implementation of such green processing methods, combined with a careful selection of biocompatible solvents, represents a promising direction for the future development of safer and more sustainable hydrogel materials [97].

The transition toward greener alternatives in binary solvent systems aligns well with the broader goals of developing more sustainable and biocompatible wearable sensing technologies. This evolution in material design not only addresses current safety concerns but also opens new possibilities for advanced biomedical applications, seamlessly connecting with the functionality and integration considerations discussed in the following section.

### 5.3. Functionality and Integration

Building upon these advances in material safety and sustainability, the integration of multiple functionalities into a single hydrogel platform has become a key research priority. The concept of sensor fusion, where various sensing modalities such as strain, temperature, and biochemical detection are combined, holds the potential to provide more comprehensive health- and environmental-monitoring capabilities. This multifunctional approach could significantly enhance the utility of wearable sensors in both clinical and consumer applications. The development of self-powered hydrogel sensors is strongly connected to their ability to perform multiple functions. The ability to generate and store energy from body movements or environmental sources is crucial for enabling long-term, continuous monitoring applications. This research direction addresses one of the primary limitations of current wearable technologies—the need for frequent recharging or battery replacement.

As the integration of multiple functionalities in binary solvent-assisted hydrogels advances, the development of robust data processing frameworks becomes increasingly critical. Beyond material properties, real-time signal processing algorithms must be optimized for handling multimodal sensor data, particularly when monitoring multiple physiological parameters simultaneously. The complexity of these integrated systems necessitates sophisticated machine learning protocols for pattern recognition and physiological monitoring, enabling more accurate analyses of biomechanical movements and health indicators. The practical implementation of these advanced materials requires systematic approaches to signal processing and system integration. Standardized calibration methods are essential for maintaining measurement accuracy across varying environmental conditions, while data fusion techniques enable comprehensive analyses of multiple sensor inputs. The development of energy-efficient signal processing architectures will be particularly crucial for enabling long-term operation in wearable applications, balancing the need for rapid response times with power consumption constraints. The synergy between material innovation and signal processing advancement represents a key enabler for next-generation sensing applications. As material properties continue to improve, parallel development of computational capabilities will enhance the overall system performance, particularly in applications requiring real-time feedback such as human–machine interfaces and continuous health monitoring. This integrated approach to material and signal processing development will be essential for realizing the full potential of binary solvent-assisted hydrogel sensors in practical applications.

As hydrogel-based sensors move towards more intimate contact with the human body, ensuring their biocompatibility and effective biointegration becomes paramount. Long-term safety studies for prolonged skin contact or potential implantation are essential. Additionally, the development of biodegradable sensors aligns with growing concerns about electronic waste and opens up new possibilities in transient electronics. Improving the interface between hydrogel sensors and biological tissues remains a critical challenge, as it directly impacts signal quality and user comfort in long-term use scenarios.

The pursuit of enhanced sensitivity and specificity in hydrogel-based sensors is ongoing. Researchers are exploring novel sensing mechanisms and material optimizations to achieve lower detection limits and higher responsivity. Developing hydrogels with higher selectivity towards target analytes is crucial for accurate biochemical sensing in complex biological fluids. This improved sensing performance is essential for the widespread adoption of hydrogel sensors in clinical and consumer applications.

As these sensors generate increasingly complex and voluminous data, the development of sophisticated data integration and analytics capabilities becomes crucial. Leveraging artificial intelligence and machine learning algorithms for processing and analyzing sensor data could lead to more accurate health predictions and personalized recommendations, potentially revolutionizing personalized healthcare and environmental monitoring.

### 5.4. Practical Implementation Challenges

The transition of hydrogel-based sensors from laboratory prototypes to mass-produced consumer devices presents a suite of practical challenges. Chief among these is the development of scalable manufacturing processes that can maintain the precise control over hydrogel properties achieved in laboratory settings. This scaling up must be achieved while simultaneously finding ways to reduce material and production costs without compromising performance, a balancing act crucial for commercial viability. Integral to large-scale production is the establishment of effective quality control measures. Ensuring consistent performance across devices is essential for building consumer trust and meeting regulatory standards. This challenge is closely linked to the need for standardization and calibration in the field. Developing robust calibration methods and standardized testing protocols for hydrogel-based sensors is necessary for their reliable use across various applications and to ensure comparability between different devices and studies.

The transition from laboratory-scale production to industrial manufacturing presents several interconnected challenges that must be addressed systematically [22,23]. Current laboratory-scale synthesis typically yields small quantities (approximately 10–50 cm^2^) of hydrogel materials with precisely controlled properties. However, scaling up to industrial production (targeting >1 m^2^ continuous sheets) while maintaining consistent performance requires significant process optimization [51,52]. Key manufacturing parameters including solution concentration, mixing conditions, crosslinking density, and curing time must be carefully controlled across larger volumes to ensure uniform material properties. Cost considerations present another critical challenge in commercialization [53,54]. Current laboratory-scale production costs, primarily driven by high-purity reagents and specialized processing conditions, typically exceed USD 100 per cm^2^. For commercial viability, production costs must be reduced to below USD 10 per cm^2^ while maintaining performance specifications. This requires both materials optimization—such as identifying lower-cost alternatives for expensive components like ionic liquids [63,64]—and process refinement to improve production efficiency and yield. The long-term stability and durability of these materials under real-world conditions require particular attention [49,59]. While laboratory demonstrations often show impressive performance over hundreds of cycles, commercial applications demand reliability over thousands of cycles spanning months or years of continuous operation. Environmental stability testing under various temperature and humidity conditions, mechanical durability assessment under repeated deformation, and accelerated aging studies are essential for validating real-world applicability. Additionally, the development of standardized testing protocols and quality control metrics is crucial for ensuring consistent performance across production batches. These practical considerations directly influence material design and processing strategies. For instance, the selection of binary solvent systems must balance performance requirements with cost and availability. Manufacturing protocols must be designed with scalability in mind, potentially requiring modifications to currently used laboratory techniques. Quality control methods must be developed to efficiently assess key performance metrics across large production volumes.

As wearable sensors become more prevalent and collect increasingly sensitive health data, ensuring data security and privacy becomes a critical concern. Robust data encryption and protection measures must be developed and implemented to safeguard user information, a challenge that requires collaboration between materials scientists, hardware engineers, and cybersecurity experts.

Finally, as the wearable technology market grows, addressing environmental concerns becomes increasingly important. Developing eco-friendly synthesis methods, exploring recycling strategies for used hydrogel sensors, and conducting comprehensive life cycle analyses are crucial steps towards minimizing the environmental impact of these technologies. This focus on sustainability must be balanced with performance requirements, presenting a complex challenge for researchers and manufacturers alike.

While binary solvent-assisted hydrogel composites show great promise for wearable sensing applications, addressing these interrelated challenges in material advancements, functionality integration, and practical implementation is crucial for realizing their full potential. Future research efforts will require interdisciplinary collaboration, combining expertise from materials science, chemistry, electronics, data analytics, and biomedical engineering to overcome these hurdles and capitalize on the emerging opportunities in this dynamic field.

## Figures and Tables

**Figure 3 materials-17-05535-f003:**
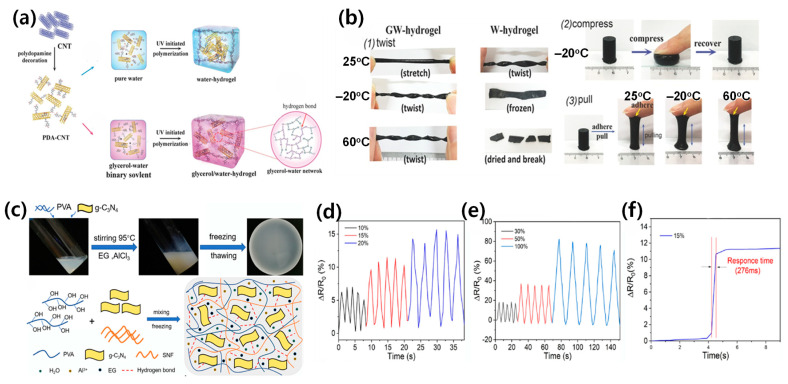
Characteristics and performance of binary solvent hydrogels. (**a**) Design of mussel-inspired glycerol/water binary solvent hydrogel. (**b**) Mechanical properties of GW hydrogel at various temperatures. (Reproduced with permission from Ref. [25]. Copyright 2018 John Wiley and Sons). (**c**) Preparation and internal structure of conductive PVA/SNF/CN organohydrogel. (**d**–**f**) Strain-sensing performance of PVA/SNF/CN organohydrogel: (**d**) small strain, (**e**) large strain, and (**f**) response time at 15% strain. (Reproduced with permission from Ref. [59]. Copyright 2021 Elsevier).

**Figure 4 materials-17-05535-f004:**
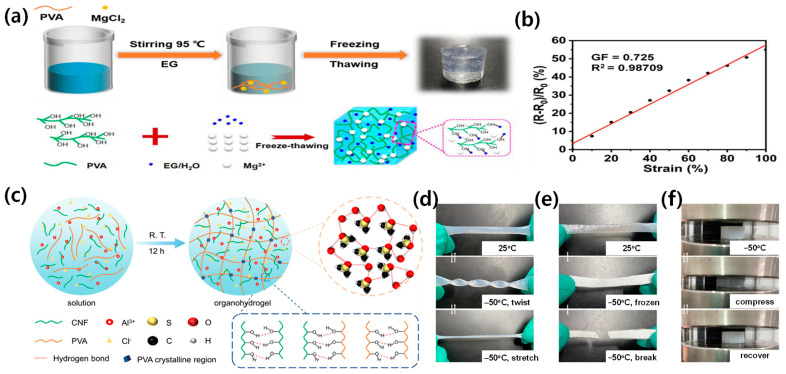
Characteristics and performance of binary solvent hydrogels. (**a**) Internal structure of PVA/EG conductive hydrogel. (**b**) Resistance changes in 25%PVA/EG hydrogel under strain. (Reproduced with permission from Ref. [42]. Copyright 2023 Elsevier). (**c**) Structure of PVA-CNF organohydrogels. (**d**–**f**) Mechanical properties of organohydrogels at −50 °C: (**d**) bending and stretching, (**e**) fracture resistance, and (**f**) shape recovery after compression. (Reproduced with permission from Ref. [49]. Copyright 2022 Elsevier).

**Figure 5 materials-17-05535-f005:**
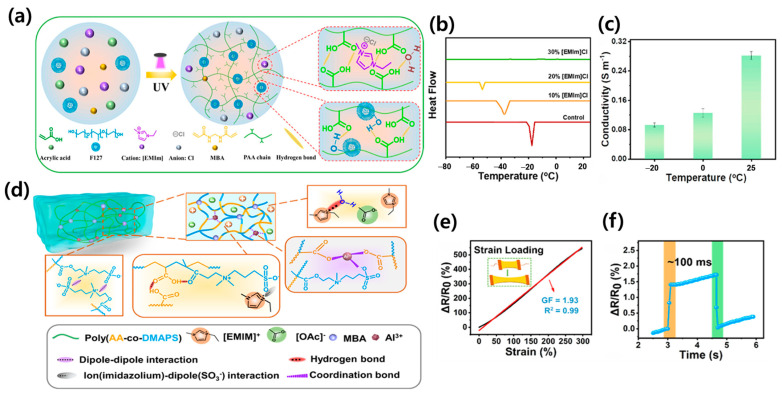
Characteristics of ionic and ionohydrogels. (**a**) Preparation process of ionic hydrogel. (**b**) DSC analysis of ionic hydrogel with varying [EMIm]Cl content. (**c**) Conductivity of ionic hydrogel at different temperatures. (Reproduced with permission from Ref. [63]. Copyright 2024 American Chemical Society). (**d**) Structure and bonding interactions in ionohydrogels. (**e**) Strain–resistance relationship of ionohydrogel-based sensors. (**f**) Response and recovery time of ionohydrogel strain sensor at 1% strain. (Reproduced with permission from Ref. [64]. Copyright 2022 American Chemical Society).

**Figure 6 materials-17-05535-f006:**
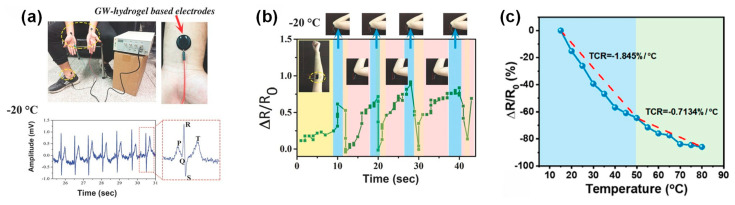
Health-monitoring performance of binary solvent hydrogels in extreme conditions. (**a**) ECG (electrocardiogram) signals detected by GW hydrogel electrodes at −20 °C. (**b**) Strain sensing of arm motions by GW hydrogel at −20 °C after one day of cooling. (Reproduced with permission from Ref. [25]. Copyright 2018 John Wiley and Sons). (**c**) Temperature sensitivity of ionic hydrogel sensor. (Reproduced with permission from Ref. [63]. Copyright 2024 American Chemical Society).

**Figure 7 materials-17-05535-f007:**
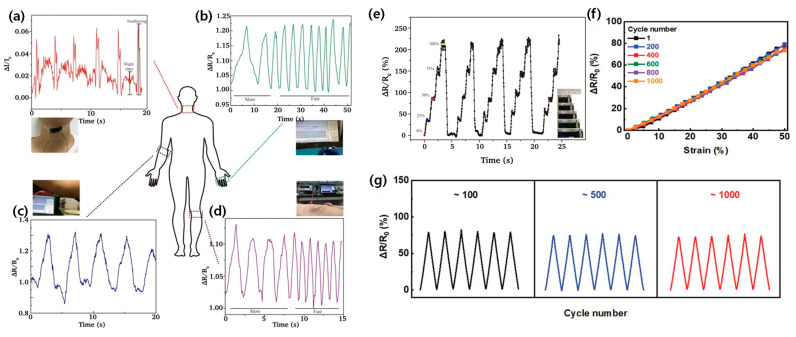
Motion capture and biomechanics performance of hydrogel-based sensors. (**a**–**d**) Relative current/resistance variations for different body movements: (**a**) throat during water drinking, (**b**) index finger, (**c**) elbow, and (**d**) knee bending. (**e**) Resistance variation in PGA gel under cyclic stretching. (Reproduced with permission from Ref. [27]. Copyright 2018 American Chemical Society). (**f**) Resistance–strain curves over multiple cycles. (**g**) Stretching cycles at 50% maximum strain. (Reproduced with permission from Ref. [53]. Copyright 2022 John Wiley and Sons).

**Figure 8 materials-17-05535-f008:**
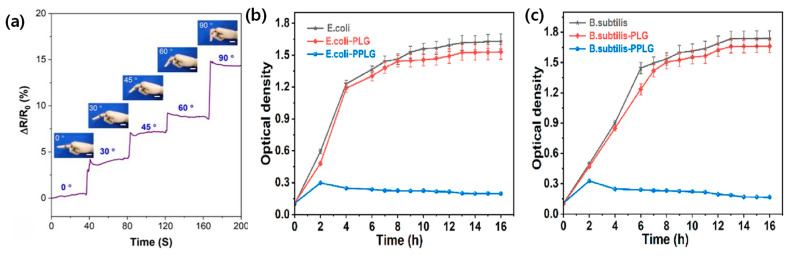
Sensing and antibacterial properties of hydrogels. (**a**) Resistance variation in G_0.3_W_0.7_ gel sensors to finger bending at different angles at −20 °C. (Reproduced with permission from Ref. [39]. Copyright 2020 American Chemical Society). (**b**,**c**) Optical density measurements showing antibacterial activity of PLG and PPLG hydrogels against (**b**) *E. coli* and (**c**) *B. subtilis*. (Reproduced with permission from Ref. [76]. Copyright 2023 Elsevier).

**Figure 9 materials-17-05535-f009:**
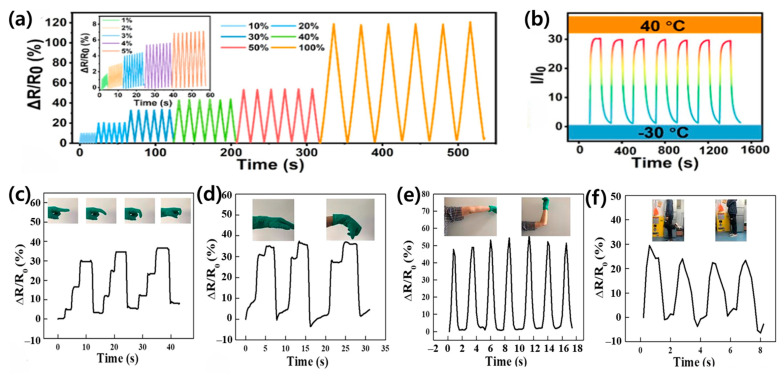
Performance of ionohydrogel-based sensors. (**a**) Resistance changes under various strains (0–300%). (**b**) Current variation in temperature sensors between −30 °C and 40 °C. (Reproduced with permission from Ref. [64]. Copyright 2022 American Chemical Society). (**c**–**f**) Motion detection: (**c**) finger bending, (**d**) wrist movement, (**e**) elbow bending, and (**f**) knee flexion. (Reproduced with permission from Ref. [77]. Copyright 2022 Elsevier).

**Figure 10 materials-17-05535-f010:**
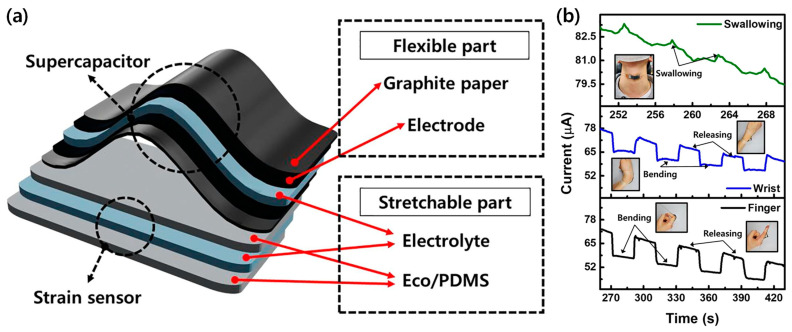
Integrated organohydrogel-based device for sensing and energy storage. (**a**) Schematic of device combining supercapacitor and strain sensor. (**b**) Bio-signal detection of swallowing, wrist bending, and finger bending at room temperature. (Reproduced with permission from Ref. [54]. Copyright 2022 Elsevier).

**Figure 11 materials-17-05535-f011:**
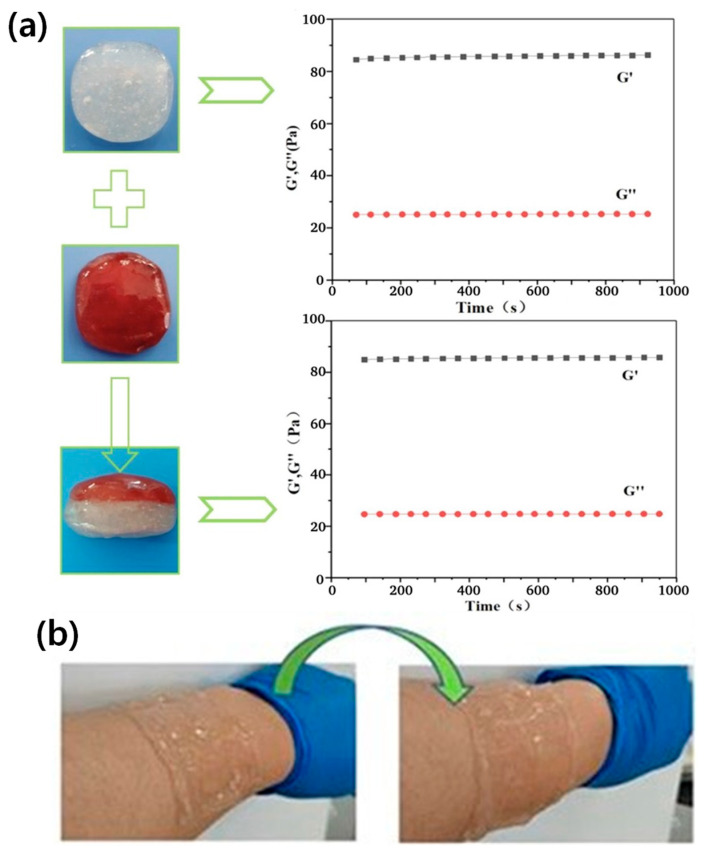
Self-healing hydrogel properties. (**a**) Visual demonstration of hydrogel mixing and corresponding rheological properties (G′ and G′′). (**b**) Application of self-healing hydrogel on skin, showing adhesion and conformability. (Reproduced with permission from Ref. [26]. Copyright 2018 American Chemical Society).

**Figure 12 materials-17-05535-f012:**
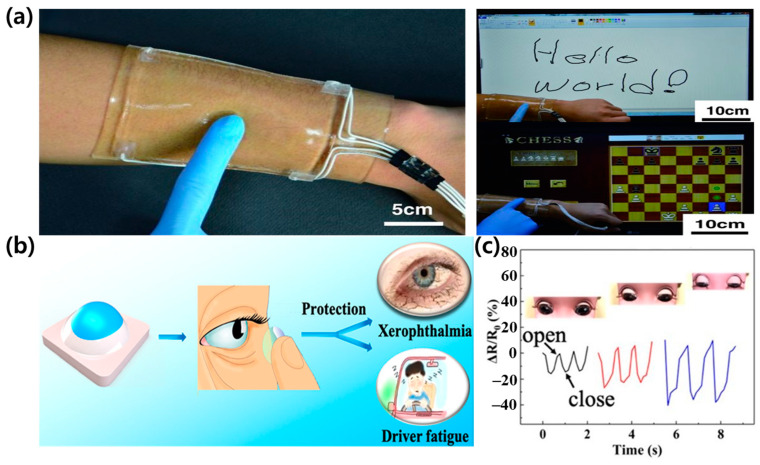
VR/AR applications of hydrogel-based sensors. (**a**) Smart touch panel for entertainment. (Reproduced with permission from Ref. [91]. Copyright 2016 American Association for the Advancement of Science). (**b**) Smart contact lenses for AR/VR and health monitoring. (**c**) Resistance changes detecting different degrees of eye blinking. (Reproduced with permission from Ref. [78]. Copyright 2020 American Chemical Society).

**Table 2 materials-17-05535-t002:** Performances of wearable sensing devices made from binary solvent-assisted hydrogel composites.

Section	Materials	Applications	Performances	Ref.
Section 3.1	PDA-CNT GW hydrogel	Self-adhesive electrocardiogram (ECG) electrodes	Electrocardiogram (ECG) signal detection Accurately and stably recorded ECG signals even at −20 °C Clear distinction of P, QRS, and T waves Strain sensor Detected arm movements after cooling at −20 °C for a day Accurately measured resistance changes corresponding to elbow joint bending LED circuit connection Maintained conductivity and adhesiveness under various environmental conditions	[25]
	Ionic hydrogel	Monitoring body temperature	Strain sensing Gauge factor (GF): 1.18 (0–200% strain), 2.15 (200–550% strain) Response time: 134 ms Stable performance over 500 cycles at 100% strain Temperature sensing −1.845%/°C (10–50 °C range) −0.713%/°C (50–80 °C range) Maintained functionality from −20 °C to 80 °C	[63]
Section 3.2	^1^ PGA hydrogel	Biosensor	Strain sensing Gauge factor (GF): 2.14 Linear response up to 100% strain Operational pressure range: 0–71.3 kPa Response time: 230 ms Stable performance over 540 cycles at 25% strain	[27]
	PANI-NP hybrid hydrogels	Human-motion interface monitoring, soft robotic applications	Self-healing Efficiency: 93.3% after 10 min Electrical healing time: 140 ms Sensing performance Gauge factor: 1.46 (0–50% strain), 2.43 (50–200% strain) Response time: 140 ms Pressure sensitivity: 0.18 kPa−1 (0–1 kPa range) Minimum detectable strain and pressure: 0.5% and 25 Pa Stable performance over 1000 cycles at 50% strain Maintained flexibility at −28 °C	[53]
Section 3.3	Conductive nanocomposite hydrogel	Wearable sensors in extremely low-temperature environments	Sensing performance Gauge factor: 0.4 (0–100% strain), up to 2.23 (600% strain) Response time: ~0.8 s Detection limit: 30° finger-bending angle Temperature tolerance Maintained flexibility and conductivity from −20 °C to 45 °C Freezing point: −43.7 °C (cooling), −21.8 °C (heating) for G_0.3_W_0.7_ gel Healing efficiency: 84.8% for G_0.2_W_0.8_ gel after 12 h Thermoresponsiveness LCST: 41.9 °C for G_0.3_W_0.7_ gel Conductivity change: 2.06 × 10^−4^ to 5.48 × 10^−4^ S/cm from 25 °C to 45 °C Optical properties Transparency: 86.2% at 610 nm for G_0.3_W_0.7_ gel	[39]
	^2^ PPLG hydrogel	Monitoring disease rehabilitation training, antibacterial wearable sensors	Sensing performance Gauge factor (GF): 1.9 (0–50% strain), 2.8 (50–175% strain), 3.6 (175–240% strain) Response time: 142 ms Recovery time: 168 ms Detectable strain range: 0–240% Maintained flexibility and conductivity from −20 °C to room temperature Inhibition zone radius: 2.6 mm (*E. coli*), 3.1 mm (*B. subtilis*) Sterilization rate: 99.5% for both *E. coli* and *B. subtilis* Retained 80% of original mass after 1-week exposure to ambient conditions	[76]
Section 3.4	Ionohydrogels	Soft wearable ionotronics	Strain sensing Gauge factor: 1.93 (0–300% strain range) Response time: <100 ms Temperature sensing Temperature coefficient of resistance (TCR): −0.035 to −0.44 °C^−1^ Temperature sensing range: −30 to 40 °C Response time: <30 s Adhesion strength8.9 kPa for paper, 5.3 kPa for copper, 5.6 kPa for PDMS, and 3.7 kPa for pig skinMaintained flexibility and conductivity from −30 °C to 40 °C Triboelectric nanogenerator performance V_OC_~48 V, I_SC_~4 μA, maximum power density~90.31 mW/m^2^ Maintained performance for 30 days in open air	[64]
	^3^ PADL hydrogel	Wearable electronic devices	Adhesive strength: 41 kPa on glass, 24 kPa on pigskin for PADL3 Maintained flexibility and conductivity at −40 °C Water retention: 65.5% of original weight after 2 weeks Antibacterial ratio: >99% against *E. coli*, *C. albicans*, and *S. aureus* Self-healing efficiency: >98% conductivity recovery after 12 h Strain sensing Gauge factor: 3.97 (0–164% strain), 7.91 (>164% strain) Working range: 0–500% strain Detection of human motion (finger, wrist, elbow, knee movements)	[77]
Section 3.5	PVA/^4^ P(AMPS-co-AAM) hydrogel	Supercapacitor	Supercapacitor Capacitance: 147.0 F/g at 25 °C, 123.4 F/g at −20 °C, 156.2 F/g at 80 °C Energy density: 13.0 Wh/kg (maximum) Power density: 3312.0 W/kg (maximum) Temperature range: −20 °C to 80 °C Cycling stability: 88% capacitance retention after 10,000 cycles at 25 °C, 94% capacitance retention after 10,000 cycles at −20 °C, 75% capacitance retention after 5000 cycles at 80 °C Bending performance: 98% capacitance retention after 1000 bending cycles at 0.19 cm bending radius Strain sensor Gauge factor: 1.77 at 25 °C, 1.61 at −20 °C, 1.50 at 80 °C Stretchability: Up to 100% strain Temperature range: −20 °C to 80 °C Detectable strain: finger bending (23% (30° angle), 42% (60° angle)), wrist bending (8%), swallowing (4%) Integrated device (supercapacitor + strain sensor) Stretchability: Up to 50% strain Temperature range: −20 °C to 80 °C Detecting bio-signals (swallowing, wrist bending, finger bending) using stored energy from the supercapacitor	[54]
Section 3.6	^5^ GG–glycerol hydrogel	Self-adhesive hydrogel dressing	Self-healing: Ultra-fast self-healing within 15 s Adhesion strength: ~2.5 kPa adhesion to copper sheet Temperature tolerance: remaining soft and elastic at −16 °C Freezing point lowered from −13.8 °C (conventional GG hydrogel) to −18.7 °C Strain sensing Detecting finger-bending motions Good repeatability in resistance change during bending cycles Water absorption: ~25% swelling ratio after 10 h in saline solution Maintained gel structure for ~36 h in water	[26]
Section 3.7	PVA glycerol/water hydrogel	Smart contact lens	Strain sensing Gauge factor: 1.56 Linear sensing range: 0–355% strain Response time: 250 ms Recovery time: 250 ms Durability: Stable over 1000 cycles at 50% strain Temperature tolerance: −20 °C to 80 °C Water vapor transmission rate: 0.0842 ± 0.0062 g/(cm^2^ day) Oxygen permeability: (21.17 ± 0.30) × 10^−11^ (cm^2^/s)(mLO_2_/mL × mmHg)	[78]

^1^ PGA: poly(vinyl alcohol)/glycerol/polyaniline; ^2^ PPLG: Polyvinyl alcohol-Polyhexamethylene biguanide hydrochloride-LiCl-Glycerol; ^3^ PADL: organohydrogel consisting of lignosulfonate nanoparticle (nano-LGS)-doped poly(acrylic acid-co-2-(methacryloyloxy)ethyl trimethyl ammonium chloride)**;**
^4^ P(AMPS-co-AAM): poly(2-acrylamido-2-methyl-1-propanesulfonic acid-co-acrylamide); ^5^ GG: guar gum.

## Data Availability

No new data were created.

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
