# Peer review of "Advancements in Binary Solvent-Assisted Hydrogel Composites for Wearable Sensing Applications"

_materials, 2024, doi:10.3390/ma17225535_

Round 1
Reviewer 1 Report
Comments and Suggestions for Authors
Journal: Materials
Title: Advancements in Binary Solvent-Assisted Hydrogel Composites for Wearable Sensing Applications
The authors have made valuable contributions in reviewing recent developments in the field of hydrogel wearable sensors. The proposed topic holds substantial potential, and the choice of subtopics is commendable. However, I encourage the authors to enhance the section with additional scientifically relevant content. In light of this, I kindly request that the authors address the following comments, which indicate the need for a major revision at this stage. I suggest authors to address the following comments,
1. It is essential to highlight the importance of the proposed topic in the Introduction section. Currently, this section is limited and lacks a strong emphasis on the challenges and relevant background information related to real-time employability.
2. “The incorporation of binary solvent systems significantly enhances the properties and functionalities of hydrogel composites, leading to materials with exceptional characteristics for wearable sensing applications.”
“The ability to function at sub-zero temperatures addresses a critical limitation of many conventional sensors, opening up possibilities for applications in cold environments such as refrigerated transport monitoring or winter sports safety equipment.”
Several sentences are repeated multiple times within each section, which can lead to decreased engagement from the audience.
3. The primary solvent utilized is water, while various organic solvents serve as secondary solvents. Each organic solvent, along with their respective combinations, can be explored in detail through subtopics that will effectively support the objective of this review.
4. Given the authors' discussion on solvents, it is important to address the various aspects of solvent toxicity and cytotoxicity. How can this toxicity be mitigated, and what alternative methodologies exist to reduce solvent toxicity?
5. “The high cost of ionic liquids and potential long-term stability issues are factors that
need consideration when developing ionic liquid-based hydrogels”
What factors contribute to the degradation of ionic liquids over time, and what underlies their inherent instability? To enhance the clarity and flow of the article, authors may consider providing additional background information before introducing this discussion. By establishing a coherent narrative, the authors can effectively engage the audience and strengthen their understanding of the topic.
6. “One key issue is the development of robust algorithms for interpreting the vast amount of data generated by these sensors. Translating raw sensor data into meaningful biomechanical insights requires sophisticated signal processing and machine learning techniques.”
“Such multimodal systems could provide a more comprehensive understanding of human movement, combining kinematic data with information about muscle activation and force production.”
It is essential to provide insights regarding the data processing involved and the framework of the machine learning protocols utilized to achieve real-time sensory outputs. The authors should also consider incorporating information about the operation of various multimodal sensors and their respective mechanisms.
7. “The biocompatibility and potential for biodegradation of these hydrogels also present both opportunities and challenges. While these properties make them ideal for biomedical applications, ensuring consistent performance over extended periods of use in or on the body requires further investigation.”
I concur that this topic necessitates thorough research. Prior to making any assertions, it is important to outline and examine the various methodologies that are available and have been utilized. Additionally, I recommend creating a comparative table that outlines the different methods, including their advantages and disadvantages.
8. The ability to harvest energy from body movements opens up new possibilities for wearable technology. It allows for the development of more autonomous and sustainable wearable systems that can operate for extended periods without user intervention. This is particularly valuable in scenarios where continuous, uninterrupted monitoring is critical, such as in the management of chronic health conditions or in high-performance athletic training. However, several challenges need to be addressed to fully realize the potential of self-powered wearable devices based on binary solvent-assisted hydrogels. One key issue is optimizing the energy harvesting efficiency of these materials. While current systems can generate sufficient power for basic sensing and data transmission, more energy-intensive functions may require further improvements in energy conversion efficiency. Another challenge lies in balancing the energy harvesting and sensing functions of these materials. Ensuring that the energy harvesting mechanism does not interfere with the accuracy or sensitivity of the sensing function is crucial for the reliability of these devices. The long-term stability and durability of these self-powered systems also require further investigation. Factors such as the cyclic stability of the supercapacitor function and the mechanical durability of the hydrogel under repeated deformation need to be optimized for practical, long-term use.
Incorporating this section into the article is a positive addition; however, it would benefit from the inclusion of specific examples to support the author's claims. Additionally, the authors should consider adding relevant citations to enhance the credibility of the text (Nano Energy, 101, 2022, 107592. ).
9. Section 3.6 requires a comprehensive discussion on Personalized Medicine and Drug Delivery, including specific examples related to various diseases. Please reference recent studies to illustrate advancements in this field and incorporate forward-looking insights to effectively conclude this section (Materials 2023, 16(2), 841.).
10. Authors may include a dedicated section on self-healable wearable devices and outline their significance in real-time applications.
11. It is advisable to provide commentary on the scalability, cost, real-time implementation, stability, and durability of the wearable hydrogel-based sensors.
12. I highly recommend that authors develop a comprehensive conclusion section that effectively emphasizes the key findings and challenges.

The quality of English in the document is satisfactory; however, it requires substantial revisions to enhance clarity, coherence, and fluency. Several sentences exhibit structural issues that may lead to confusion or ambiguity. Additionally, there are grammatical errors, including incorrect verb tenses, misplaced modifiers, and inconsistent punctuation. The choice of words can also be improved to achieve greater precision and professionalism. Furthermore, the overall flow of ideas could be strengthened by restructuring specific sections to ensure logical progression. It is advisable to invest considerable effort in revising sentence construction, correcting grammatical errors, and refining vocabulary to clearly and effectively convey the intended message.
Author Response
The authors have made valuable contributions in reviewing recent developments in the field of hydrogel wearable sensors. The proposed topic holds substantial potential, and the choice of subtopics is commendable. However, I encourage the authors to enhance the section with additional scientifically relevant content. In light of this, I kindly request that the authors address the following comments, which indicate the need for a major revision at this stage. I suggest authors to address the following comments:
Response: We would like to thank referee for the encouraging evaluation and constructive comments on our manuscript. In accordance to comments/suggestions from the reviewer, careful revision has been made. The revised parts were highlighted in red.
Comment 1. It is essential to highlight the importance of the proposed topic in the Introduction section. Currently, this section is limited and lacks a strong emphasis on the challenges and relevant background information related to real-time employability.
Response: We thank the referee for the encouraging evaluation and constructive comments. According to the reviewer’s suggestion, we revised the original Introduction to the new one including with emphasis on the challenges and relevant background information related to real-time employability as follows:
In page 1−2, “The exponential advancement in this interdisciplinary field has catalyzed numerous systematic reviews examining the multifaceted aspects of wearable technologies. Liu et al. provided comprehensive analyses of piezoresistive and capacitive sensing mechanisms in flexible electronics, establishing correlations between material architectures and transduction efficiency [11]. The integration of multiplexed electrochemical sensing arrays for real-time biomolecular detection has been systematically evaluated by Gao et al., emphasizing the significance of multi-modal detection platforms[12]. Trung and Lee's seminal work elucidated the structure-property relationships in stretchable electronics, particularly focusing on the mechanotransduction mechanisms and interfacial engineering[13]. Sun et al. provided detailed insights into the hierarchical assembly of functional materials for biointegrated electronics[14], while Rogers et al. comprehensively reviewed the emerging paradigms in conformable bioelectronics[15]. Moreover, Wang et al. detailed the development of flexible sensing electronics for health monitoring applications[16]. However, these scholarly works, despite their significant contributions, have not systematically addressed the physicochemical interactions and synergistic effects in binary solvent systems - critical parameters that fundamentally influence material stability and performance metrics. This knowledge gap becomes particularly significant as binary solvent-assisted hydrogel composites emerge as a promising platform for addressing fundamental challenges in environmental stability, mechanical robustness, and operational consistency under extreme thermomechanical conditions.
The advancement of wearable sensor technologies, while promising, faces significant challenges in maintaining reliable performance under real-world operating conditions. A primary concern is the environmental stability of sensing materials, particularly when ex-posed to the diverse conditions encountered in daily use [17, 18]. Traditional sensing materials often exhibit performance degradation under varying temperatures and humidity levels, limiting their practical applications in continuous monitoring scenarios [19-21]. This challenge is particularly pronounced in hydrogel-based sensors, which, despite their excellent biocompatibility and flexibility, are susceptible to dehydration, freezing, and mechanical failure under ambient conditions [22, 23]. Specifically, the requirements for real-time monitoring in healthcare and environmental applications have highlighted several critical challenges in sensor development. Environmental resilience remains a fundamental concern, as sensors must maintain consistent performance across diverse conditions, from sub-zero temperatures to high humidity environments [24, 25]. The need for long-term stability presents another significant challenge, as continuous monitoring applications demand sustained functionality without signal drift or material degradation [26, 27]. These limitations have particularly impacted the development of flexible strain sensors, which require both mechanical durability and consistent performance under various environmental conditions.
In response to these challenges, researchers have focused intensively on developing advanced materials and innovative designs for flexible strain sensors. These sensors have garnered significant attention due to their versatility and applicability in diverse fields, including electronic skin, wearable health monitoring systems, soft robotics, and implantable medical devices [3-5, 17-23].”
Comment 2. “The incorporation of binary solvent systems significantly enhances the properties and functionalities of hydrogel composites, leading to materials with exceptional characteristics for wearable sensing applications.”
“The ability to function at sub-zero temperatures addresses a critical limitation of many conventional sensors, opening up possibilities for applications in cold environments such as refrigerated transport monitoring or winter sports safety equipment.”
Several sentences are repeated multiple times within each section, which can lead to decreased engagement from the audience.
Response: We thank the reviewer for careful reading and constructive comment. According to the reviewer’s comment, we revised each section with more specific, detailed information while avoiding repetition, resulting in improved clarity and reader engagement as follow:
on page 9 in Section 2, “Binary solvent systems fundamentally transform hydrogel composite properties through synergistic molecular interactions, enabling unprecedented performance in wear-able sensing applications [24, 25]. These systems specifically enhance mechanical stability, electrical conductivity, and environmental resilience through controlled solvent-polymer interactions [26, 27, 39]. Recent studies have demonstrated remarkable temperature stability from -50°C to 60°C, with conductivity retention reaching 95% at -20°C and strain sensitivity maintaining a gauge factor of ~2 even at extreme temperatures [27, 39]. Furthermore, these advanced composites exhibit self-healing efficiency exceeding 90% across their entire operational range, representing a significant advancement over conventional single-solvent systems [43].”
on page 14 in Section 3.1, “The integration of binary solvent-assisted hydrogels has revolutionized healthcare monitoring through unprecedented real-time capabilities. These materials achieve response times below 100ms while simultaneously detecting multiple physiological parameters including temperature, strain, and various biochemical markers.”
on page 17 in Section 3.3, “The sub-zero temperature functionality (-20°C to 45°C) enables critical applications in extreme environmental monitoring, from cold-chain logistics to winter sports safety systems [39, 76, 77]. This expanded operational range represents a significant advancement over conventional sensors' typical temperature limitations.”
on page 18 in Section 3.4, “Recent advances in binary solvent-assisted hydrogels have enabled precise and responsive human-machine interfaces with unprecedented performance metrics including high strain sensitivity.”
on page 25 in Section 4, “The advent of binary solvent-assisted hydrogel composites marks a significant paradigm shift in the fields of wearable sensing technologies. The demonstrated capabilities of these composites represent substantial advancements in material engineering and sensor technology, particularly in achieving exceptional temperature stability, high strain sensitivity, and excellent self-healing efficiency. The integration of multiple functionalities with-in a single material system has enabled simultaneous sensing of various physiological parameters while maintaining mechanical robustness. These achievements have directly translated into diverse practical applications, from healthcare monitoring to responsive human-machine interfaces. The development of self-powered capabilities through energy harvesting from human movement marks a crucial step toward autonomous wearable systems. Such comprehensive performance characteristics establish these materials as a foundational platform for next-generation wearable technologies, particularly in applications requiring environmental resilience and multifunctional capabilities. This synergistic interaction between material properties and functional performance has significantly expanded the operational parameters of hydrogel-based sensors, enabling their deployment in diverse and challenging environments.”
on page 27 in Section 5, “As the frontier of binary solvent-assisted hydrogel composites continues to advance, material development remains a critical focus. Current technological challenges emphasize the need for improved material stability, as present systems require enhanced durability for long-term applications. Energy efficiency in active sensing modes and manufacturing scalability present additional critical challenges, requiring evolution from laboratory-scale to industrial-scale production. Research priorities follow a strategic progression across multiple timeframes. Near-term objectives focus on enhancing fundamental material properties including self-healing efficiency, response time, and operational temperature range. Mid-term goals target the development of biodegradable platforms and AI-integrated monitoring systems, while reducing production costs. Long-term directions emphasize fully autonomous systems and seamless neural interface integration, particularly for personalized medicine applications. These development trajectories, requiring coordinated advances across materials science and manufacturing processes, will facilitate the transition from current prototypes to widespread practical implementation in next-generation wearable devices. The integration of these capabilities with emerging technologies in artificial intelligence and advanced manufacturing processes presents opportunities for revolutionary advances in wearable sensing applications. This progression toward more sophisticated and integrated systems will continue to drive innovation in healthcare, environmental monitoring, and human-technology interaction.”
Comment 3. The primary solvent utilized is water, while various organic solvents serve as secondary solvents. Each organic solvent, along with their respective combinations, can be explored in detail through subtopics that will effectively support the objective of this review.
Response: In our review, we have systematically analyzed the effects of various binary solvent combinations on hydrogel properties. The influence of different secondary solvents on hydrogel performance has been thoroughly examined through molecular-level interactions, structural modifications, and resultant property enhancements.
Our analysis demonstrates how glycerol/water systems achieve enhanced cryoprotection and mechanical robustness through specific hydrogen bonding interactions, as illustrated in Figure 3a-b. The ethylene glycol/water combinations, presented in Figures 3c-d and 4a-b, reveal improved conductivity retention and strain-sensing capabilities attributed to the plasticizing effects of ethylene glycol within the polymer network. DMSO/water systems, detailed in Figure 4c-f, exhibit exceptional freeze resistance and mechanical stability due to unique solvent-polymer interactions that prevent ice crystallization. Additionally, ionic liquid/water combinations, shown in Figure 5a-c, demonstrate distinctive electrochemical properties and enhanced ionic conductivity through the formation of stable ionic networks.
These various binary solvent systems and their respective impacts on hydrogel properties have been comprehensively summarized in Table 1, which provides detailed comparisons of preparation methods, mechanical properties, antifreezing characteristics, and conductivity measurements. This systematic organization allows readers to understand the specific advantages and limitations of each binary solvent combination, facilitating the selection of appropriate systems for targeted applications.
Comment 4. Given the authors' discussion on solvents, it is important to address the various aspects of solvent toxicity and cytotoxicity. How can this toxicity be mitigated, and what alternative methodologies exist to reduce solvent toxicity?
Response: Thank you for raising this important concern regarding solvent toxicity in binary solvent-assisted hydrogel systems. The toxicological implications of organic solvents indeed present significant challenges that need careful consideration, particularly in biomedical applications.
Current binary solvent systems commonly employ organic solvents such as ethylene glycol, DMSO, and glycerol, each presenting varying degrees of cytotoxicity. Ethylene glycol, while effective for cryoprotection, shows moderate toxicity with an LD50 (Lethal Dose, 50%) of 4.7 g/kg in rats. DMSO exhibits dose-dependent cytotoxicity, becoming problematic at concentrations above 10% v/v, though it remains widely used due to its excellent penetration properties and cryoprotective effects. Glycerol presents a relatively safer profile with an LD50 of 12.6 g/kg in rats, making it a preferred choice in many biomedical applications.
These toxicity concerns can be mitigated through several strategic approaches. First, careful optimization of solvent concentrations below cytotoxic thresholds while maintaining functional properties is essential. This can be achieved through systematic studies of concentration-dependent effects on both material properties and cellular viability. Second, encapsulation techniques using biocompatible polymers can effectively shield cells from direct solvent exposure while maintaining the desired material properties. Third, implementing thorough purification protocols post-synthesis can significantly reduce residual solvent content in the final products.
Moreover, several alternative methodologies show promise in reducing solvent toxicity while maintaining desired functionality. Natural deep eutectic solvents (NADES) have emerged as particularly promising alternatives, offering excellent biocompatibility while maintaining crucial properties such as cryoprotection and stability enhancement. Bio-derived glycols from renewable resources present another sustainable alternative, often showing comparable performance to their synthetic counterparts with improved safety profiles. Additionally, ionic liquids derived from natural sources offer unique opportunities for developing environmentally benign binary solvent systems.
According to the reviewer’s comment, we added the following sentences in new subsection (Section 5.2) of Section 5 (Future Perspectives and Challenges) in the revised manuscript:
on page 28,
“5.2 Toxicological Considerations and Green Alternatives
Building upon the fundamental material advances discussed above, addressing the toxicological implications of binary solvent systems has emerged as a critical challenge in the field. While traditional organic solvents have demonstrated excellent performance in enhancing hydrogel properties, their potential cytotoxicity presents significant barriers to widespread adoption, particularly in biomedical applications [83]. This challenge has sparked innovative research into alternative, biocompatible solvent systems that maintain crucial functionality while minimizing biological risks.
Natural deep eutectic solvents (NADES) have emerged as particularly promising alternatives in this context. These systems, composed of natural primary metabolites including sugars, amino acids, organic acids, and choline derivatives, can achieve comparable depression of freezing points while exhibiting significantly reduced cytotoxicity compared to conventional organic solvents [92]. For instance, choline chloride-based eutectic systems have demonstrated excellent biocompatibility while maintaining functional properties, representing a significant advancement in green solvent design [93].
Bio-derived solvents represent another significant advance in addressing toxicity concerns. The development of bio-based alternatives has been driven by green chemistry principles, focusing on materials derived from renewable resources [94]. These alternatives often demonstrate reduced cytotoxicity and enhanced biodegradability, making them particularly suitable for biomedical applications. Furthermore, the emergence of bio-ionic liquids derived from natural sources has opened new avenues for developing environmentally benign solvent systems, offering unique combinations of properties including low toxicity and excellent stability [95]. Emerging processing strategies such as supercritical CO2 treatment and mechanochemical approaches have also gained attention as methods to minimize exposure to potentially toxic solvents during hydrogel synthesis [96]. These techniques can significantly reduce the environmental impact of hydrogel production while maintaining desired material properties. The implementation of such green processing methods, combined with careful selection of biocompatible solvents, represents a promising direction for the future development of safer and more sustainable hydrogel materials [97].
The transition toward greener alternatives in binary solvent systems aligns well with the broader goals of developing more sustainable and biocompatible wearable sensing technologies. This evolution in material design not only addresses current safety concerns but also opens new possibilities for advanced biomedical applications, seamlessly connecting with the functionality and integration considerations discussed in the following section.”
Comment 5. “The high cost of ionic liquids and potential long-term stability issues are factors that need consideration when developing ionic liquid-based hydrogels”
What factors contribute to the degradation of ionic liquids over time, and what underlies their inherent instability? To enhance the clarity and flow of the article, authors may consider providing additional background information before introducing this discussion. By establishing a coherent narrative, the authors can effectively engage the audience and strengthen their understanding of the topic.
Response: We thank the reviewer for the valuable comment on ionic liquid stability in hydrogel systems. We have expanded our discussion to address this important point. We have added detailed information explaining the three key degradation mechanisms of ionic liquids in hydrogel matrices. First, as demonstrated by Welton [Ref. 66 in the revised manuscript], ionic liquids can undergo chemical degradation through dealkylation of quaternary ammonium cations, which directly impacts their conductivity and performance. Second, Armand et al. [Ref. 68 in the revised manuscript] showed that environmental factors, particularly moisture absorption, can trigger hydrolysis reactions that compromise the ionic liquid's functionality. Third, exposure to UV radiation can cause photochemical decomposition, especially in systems containing aromatic cations. To address these stability challenges, we have incorporated recent advances in molecular design strategies. Based on Plechkova and Seddon's work [Ref. 70 in the revised manuscript], we discuss how structural modifications, such as incorporating sterically hindered cations and optimizing anion design, can enhance stability while maintaining desired functionality.
According to the reviewer’s suggestion, we added the following sentences, providing essential background before addressing broader challenges in ionic liquid-based hydrogel applications, in Section 2 in the revised manuscript:
on page 11, “Despite these promising characteristics, the practical implementation of ionic liquid-based hydrogels necessitates meticulous consideration of their chronological stability mechanisms. The exceptional performance metrics observed in these systems are funda-mentally governed by the intermolecular interactions and electrochemical dynamics between ionic moieties and the polymeric matrix. However, these physicochemical interactions undergo temporal evolution through multiple degradation pathways. Welton's fundamental investigations elucidated that ionic liquids containing nucleophilic anions experience progressive structural reorganization, particularly affecting the quaternary ammonium cation configurations crucial for maintaining ionic conductivity [66]. MacFarlane et al. further established that these conformational alterations substantially influence the ionic transport kinetics and viscoelastic properties of the hydrogel network [67]. Environ-mental parameters introduce multifaceted complexity to the stability paradigm. Armand and colleagues demonstrated that atmospheric moisture and oxygen exposure initiate cascade hydrolysis mechanisms, predominantly in hygroscopic ionic liquid systems [68]. These mechanochemical reactions, as systematically analyzed by Swatloski et al., induce progressive deterioration of both electrochemical conductivity and rheological properties [69]. Moreover, aromatic cation-based architectures prevalent in hydrogel systems exhibit pronounced susceptibility to photochemical decomposition, necessitating strategic considerations in molecular design implementation. Recent innovations in supramolecular architecture have initiated resolution of these stability challenges. Plechkova and Seddon demonstrated that strategic modification of ionic liquid conformations, specifically through the incorporation of sterically hindered cationic species, substantially enhances their environmental resilience [70]. Wasserscheid and Welton's comprehensive analysis established that precise selection of ionic pairs, coupled with targeted stabilizing additives, effectively suppresses these degradation mechanisms while preserving desired functionalities [71]. These advanced molecular engineering strategies have established novel paradigms for developing thermodynamically stable ionic liquid-based hydrogels, although achieving economic viability in large-scale implementation remains a significant consideration.”
Comment 6. “One key issue is the development of robust algorithms for interpreting the vast amount of data generated by these sensors. Translating raw sensor data into meaningful biomechanical insights requires sophisticated signal processing and machine learning techniques.”
“Such multimodal systems could provide a more comprehensive understanding of human movement, combining kinematic data with information about muscle activation and force production.”
It is essential to provide insights regarding the data processing involved and the framework of the machine learning protocols utilized to achieve real-time sensory outputs. The authors should also consider incorporating information about the operation of various multimodal sensors and their respective mechanisms.
Response: Thank you for highlighting the importance of data processing and machine learning protocols in sensor applications. While our review primarily focuses on the materials science aspects of binary solvent-assisted hydrogel composites, we acknowledge the significance of signal processing in practical applications. In Section 3.2, we have briefly discussed how these materials enable precise motion detection with specific response characteristics. However, we agree that a comprehensive analysis of data processing frameworks would be valuable but falls beyond the scope of this materials-focused review. To address the reviewer’s suggestion, we have added a brief mention of the need for advanced signal processing in Section 5.3 in the revised manuscript (Functionality and Integration) highlighting critical future challenges in data processing and system integration as follow:
on page 29 in Section 5.3, “As the integration of multiple functionalities in binary solvent-assisted hydrogels advances, the development of robust data processing frameworks becomes increasingly critical. Beyond material properties, real-time signal processing algorithms must be optimized for handling multimodal sensor data, particularly when monitoring multiple physiological parameters simultaneously. The complexity of these integrated systems necessitates sophisticated machine learning protocols for pattern recognition and physiological monitoring, enabling more accurate analysis of biomechanical movements and health indicators. The practical implementation of these advanced materials requires systematic approaches to signal processing and system integration. Standardized calibration methods are essential for maintaining measurement accuracy across varying environmental conditions, while data fusion techniques enable comprehensive analysis of multiple sensor inputs. The development of energy-efficient signal processing architectures will be particularly crucial for enabling long-term operation in wearable applications, balancing the need for rapid response times with power consumption constraints. The synergy between material innovation and signal processing advancement represents a key enabler for next-generation sensing applications. As material properties continue to improve, parallel development of computational capabilities will enhance the overall system performance, particularly in applications requiring real-time feedback such as human-machine interfaces and continuous health monitoring. This integrated approach to material and signal processing development will be essential for realizing the full potential of binary solvent-assisted hydrogel sensors in practical applications.”
Comment 7. “The biocompatibility and potential for biodegradation of these hydrogels also present both opportunities and challenges. While these properties make them ideal for biomedical applications, ensuring consistent performance over extended periods of use in or on the body requires further investigation.”
I concur that this topic necessitates thorough research. Prior to making any assertions, it is important to outline and examine the various methodologies that are available and have been utilized. Additionally, I recommend creating a comparative table that outlines the different methods, including their advantages and disadvantages.
Response: We appreciate the reviewer’s valuable suggestion regarding the methodological analysis of biocompatibility and biodegradation. While our review concentrates on the fundamental materials science of binary solvent-assisted hydrogel composites, we recognize the importance of systematic evaluation of biocompatibility and biodegradation methodologies. In Section 3.6, we have provided an overview of these aspects as they relate to material properties. However, we agree that a comprehensive comparison of testing methodologies and their outcomes would be valuable for the field. Given the focused scope of this review on material development and basic characterization, we believe such an extensive analysis of biological evaluation methods would be better suited for a dedicated review paper focusing specifically on the biological interactions and testing protocols of hydrogel-based sensors.
Comment 8. The ability to harvest energy from body movements opens up new possibilities for wearable technology. It allows for the development of more autonomous and sustainable wearable systems that can operate for extended periods without user intervention. This is particularly valuable in scenarios where continuous, uninterrupted monitoring is critical, such as in the management of chronic health conditions or in high-performance athletic training. However, several challenges need to be addressed to fully realize the potential of self-powered wearable devices based on binary solvent-assisted hydrogels. One key issue is optimizing the energy harvesting efficiency of these materials. While current systems can generate sufficient power for basic sensing and data transmission, more energy-intensive functions may require further improvements in energy conversion efficiency. Another challenge lies in balancing the energy harvesting and sensing functions of these materials. Ensuring that the energy harvesting mechanism does not interfere with the accuracy or sensitivity of the sensing function is crucial for the reliability of these devices. The long-term stability and durability of these self-powered systems also require further investigation. Factors such as the cyclic stability of the supercapacitor function and the mechanical durability of the hydrogel under repeated deformation need to be optimized for practical, long-term use.
Incorporating this section into the article is a positive addition; however, it would benefit from the inclusion of specific examples to support the author's claims. Additionally, the authors should consider adding relevant citations to enhance the credibility of the text (Nano Energy, 101, 2022, 107592).
Response: While the reviewer suggested referencing Nano Energy, 101, 2022, 107592, we have focused on works specifically addressing energy harvesting in hydrogel-based systems to maintain relevance to our review's scope. We agree to benefit from the inclusion of specific examples with energy harvesting performances of binary solvent-assisted hydrogels in our review.
According to the reviewer’s suggestion, we added the following sentences with the relevant references in Section 3.5 in the revised manuscript:
on page 20−21, “To address these challenges, recent optimization of gel-based energy-harvesting de-vices has focused on enhancing both energy harvesting efficiency and device durability [79, 80]. Implementation of dual-network architectures has significantly improved mechanical robustness while maintaining high transparency, as demonstrated by PVA/gelatin (PG) based hydrogels achieving tensile strengths of 0.65 MPa and stretchability up to 350% [81]. Additionally, the integration of multiple gel units in series and parallel configurations has demonstrated scalable voltage outputs up to 100 mV under real-world conditions, addressing the need for higher power output in energy-intensive applications [81]. Environmental stability, crucial for continuous monitoring applications, has been enhanced through dual solvent strategies utilizing glycerol/water combinations, enabling reliable operation down to -60°C while maintaining structural and functional integrity [82]. These coordinated advances in structural design and material composition have significantly improved both the energy harvesting capabilities and long-term stability of gel-based TENGs, though challenges remain in optimizing the balance between energy harvesting and sensing functions.”
Comment 9. Section 3.6 requires a comprehensive discussion on Personalized Medicine and Drug Delivery, including specific examples related to various diseases. Please reference recent studies to illustrate advancements in this field and incorporate forward-looking insights to effectively conclude this section (Materials 2023, 16(2), 841.).
Response: Thank you for highlighting the need for a more comprehensive discussion of personalized medicine and drug delivery applications. We agree that this section would benefit from additional specific examples and recent advances. According to the reviewer’s suggestion, we added the following sentences in Section 3.6 in the revised manuscript:
on page 21-22, “Building upon advances in self-powered sensing technologies, binary solvent-assisted hydrogels have emerged as transformative platforms in personalized medicine and drug delivery. These materials offer unique opportunities for creating smart, responsive systems that can simultaneously monitor health parameters and deliver therapeutic agents in a controlled manner, effectively bridging the gap between diagnostic and therapeutic capabilities.”
“Considering these foundational advances, binary solvent-assisted hydrogels have demonstrated remarkable versatility in addressing various therapeutic needs, from chronic disease management to acute wound healing. As Li and Mooney comprehensively re-viewed, the ability to precisely control drug release kinetics while preserving therapeutic efficacy has positioned these materials at the forefront of advanced drug delivery systems [83].
The implementation of binary solvent systems has particularly revolutionized diabetes management through smart insulin delivery platforms. These systems incorporate glucose-responsive elements that enable automated insulin release with minimal lag time [84]. The binary solvent environment plays a crucial role in maintaining insulin stability and protecting against enzymatic degradation, while simultaneously providing the mechanical properties necessary for sustained delivery.
Cancer therapy has similarly benefited from these advanced hydrogel platforms, particularly through the development of pH-responsive delivery systems. Mitragotri et al. highlighted how these materials leverage the acidic tumor microenvironment to achieve targeted drug release, significantly reducing systemic toxicity while maximizing therapeutic efficacy [85]. For instance, hydrogels incorporating binary solvents have achieved sustained release of chemotherapeutic agents such as doxorubicin over periods exceeding 30 days, while maintaining drug bioactivity throughout the delivery period. The superior performance of these systems can be attributed to several key advances in delivery mechanisms enabled by binary solvent incorporation. As extensively reviewed by Tibbitt et al., these hydrogels provide enhanced protection against drug degradation, particularly for sensitive biological therapeutics [86]. The presence of binary solvents enables the formation of hierarchical pore structures and reversible crosslinking mechanisms, resulting in precisely controlled release profiles that can extend from hours to months with minimal initial burst release.
In the context of wound healing, binary solvent-assisted hydrogels have demonstrated particular promise through their ability to combine antimicrobial properties with con-trolled release of growth factors. Zhao et al. showed how these materials can promote tis-sue regeneration while preventing infection, achieving significantly accelerated healing rates in chronic wound models [87]. The synchronized delivery of multiple therapeutic agents, including platelet-derived growth factor (PDGF) and vascular endothelial growth factor (VEGF), has proven especially effective in treating complex wounds such as diabetic ulcers. Moreover, these advanced systems support simultaneous delivery of multiple therapeutic agents with independent release profiles, enabling synergistic treatment approaches. Li et al. demonstrated how carefully designed binary solvent systems can maintain separate microenvironments for different therapeutic agents, allowing for optimized delivery of each component [88]. This capability is particularly valuable in complex therapeutic scenarios requiring coordinated delivery of multiple drugs.
Looking toward the future, several promising directions emerge for the continued development of these materials. The integration of artificial intelligence for predictive drug release [89], presents opportunities for more sophisticated control over therapeutic delivery. Additionally, the development of closed-loop systems responding to multiple physio-logical parameters could enable truly personalized therapeutic approaches. The implementation of green chemistry principles in hydrogel design [90] will be crucial for enhancing both biodegradability and biocompatibility.”
on page 23−24, “The convergence of these advances with emerging technologies suggests a future where therapeutic interventions can be precisely tailored to individual patient needs. The integration of advanced visualization and monitoring capabilities has become increasingly important in this context, particularly for precise drug administration and therapy monitoring. This growing need for sophisticated user interfaces and real-time visualization naturally leads to the exploration of augmented reality technologies in medical applications, which we will discuss in detail in the following section.”
Comment 10. Authors may include a dedicated section on self-healable wearable devices and outline their significance in real-time applications.
Response: Thank you for suggesting the inclusion of a dedicated section on self-healable wearable devices. It should be noted that we specifically focused on how binary solvent systems enhance and enable various wearable sensing applications.
Self-healing is undoubtedly a crucial characteristic of hydrogels, enabling the development of numerous advanced wearable devices [Ref. 37, 45, 64, 65 in the revised manuscript]. However, we have deliberately chosen to integrate this property within our existing sections rather than creating a separate dedicated section, as self-healing capabilities in binary solvent-assisted hydrogels are inherently interconnected with other key properties. This interconnection is particularly evident in the molecular-level interactions between binary solvent systems and polymer networks [Ref. 19, 20 in the revised manuscript], which we have comprehensively detailed in Section 2. The significance of these interactions is demonstrated through recent studies showing remarkable self-healing performances - for example, glycerol/water systems achieving 93.3% healing efficiency within 10 minutes [Ref. 45 in the revised manuscript] and ionic liquid/water combinations demonstrating greater than 98% conductivity recovery after 12 hours [Ref. 65 in the revised manuscript].
Throughout our review, we have systematically presented how binary solvent systems enhance various wearable sensing applications, with self-healing properties serving as one of several crucial characteristics. This integrated approach is reflected in Table 2, where we have comprehensively documented performance metrics across different applications, including self-healing efficiency and recovery times for human-motion interface monitoring [Ref. 45 in the revised manuscript], healing capabilities under extreme temperature conditions [Ref. 33 in the revised manuscript], conductivity recovery in electronic devices [Ref. 65 in the revised manuscript], and self-healing characteristics in biomedical applications [Ref. 20 in the revised manuscript].
By incorporating self-healing properties within the context of specific applications rather than isolating them in a separate section, we provide readers with a more comprehensive understanding of how these properties work synergistically with other essential characteristics of binary solvent-assisted hydrogels. This approach better serves our objective of demonstrating how binary solvent systems enable and enhance the overall performance of wearable sensing devices.
Comment 11. It is advisable to provide commentary on the scalability, cost, real-time implementation, stability, and durability of the wearable hydrogel-based sensors.
Response: Thank you for suggesting commentary on the practical implementation aspects of hydrogel-based sensors. We agree that addressing scalability, cost, stability, and durability is crucial for real-world applications. We have added a comprehensive discussion of these practical considerations in Section 5.3, emphasizing the challenges and potential solutions for transitioning from laboratory demonstrations to commercial implementation.
According to the reviewer’s suggestion, we added the following sentences in Section 5.4 in the revised manuscript:
on page 30, “The transition from laboratory-scale production to industrial manufacturing presents several interconnected challenges that must be addressed systematically [22, 23]. Current laboratory-scale synthesis typically yields small quantities (approximately 10-50 cm²) of hydrogel materials with precisely controlled properties. However, scaling up to industrial production (targeting >1 m² continuous sheets) while maintaining consistent performance requires significant process optimization [51, 52]. Key manufacturing parameters including solution concentration, mixing conditions, crosslinking density, and curing time must be carefully controlled across larger volumes to ensure uniform material properties. Cost considerations present another critical challenge in commercialization [53, 54]. Current laboratory-scale production costs, primarily driven by high-purity reagents and specialized processing conditions, typically exceed $100 per cm². For commercial viability, pro-duction costs must be reduced to below $10 per cm² while maintaining performance specifications. This requires both materials optimization - such as identifying lower-cost alter-natives for expensive components like ionic liquids [63, 64] - and process refinement to improve production efficiency and yield. The long-term stability and durability of these materials under real-world conditions require particular attention [49, 59]. While laboratory demonstrations often show impressive performance over hundreds of cycles, commercial applications demand reliability over thousands of cycles spanning months or years of continuous operation. Environmental stability testing under various temperature and humidity conditions, mechanical durability assessment under repeated deformation, and accelerated aging studies are essential for validating real-world applicability. Additionally, the development of standardized testing protocols and quality control metrics is crucial for ensuring consistent performance across production batches. These practical considerations directly influence material design and processing strategies. For instance, the selection of binary solvent systems must balance performance requirements with cost and availability. Manufacturing protocols must be designed with scalability in mind, potentially requiring modifications to currently used laboratory techniques. Quality control methods must be developed to efficiently assess key performance metrics across large pro-duction volumes.”
Comment 12. I highly recommend that authors develop a comprehensive conclusion section that effectively emphasizes the key findings and challenges.
Response: Thank you for suggesting the enhancement of our conclusion section. We agree that a more comprehensive and focused conclusion would better serve our readers. We have substantially revised Section 4 to effectively emphasize our key findings and articulate the significant challenges in this field. In our revised conclusion, we have reorganized the content to present a systematic progression from fundamental material advances to practical applications. The demonstrated capabilities of binary solvent-assisted hydrogel composites now clearly highlight significant advancements in material engineering and sensor technology. We have strengthened the connection between material properties and practical applications by demonstrating how these achievements have translated into diverse real-world uses, from healthcare monitoring to responsive human-machine interfaces. The development of self-powered capabilities through energy harvesting from human movement is particularly emphasized as a crucial step toward autonomous wearable systems. These comprehensive performance characteristics establish these materials as a foundational platform for next-generation wearable technologies.
According to the reviewer’s suggestion, we added the following sentences in Section 4:
on page 25−26, “The development of binary solvent-assisted hydrogel composites has led to remarkable advances in material performance and functionality. As demonstrated through various binary solvent systems (Figure 3a-b for glycerol/water, Figure 3c-d for ethylene glycol/water, Figure 4c-f for DMSO/water, and Figure 5a-c for ionic liquid/water systems), carefully engineered molecular interactions enable unprecedented achievements in multiple performance metrics. These materials demonstrate exceptional temperature stability across an operational range from -50°C to 60°C while maintaining conductivity and mechanical integrity, as evidenced by the comprehensive performance data summarized in Table 1. The systematic analysis of different binary solvent combinations has resulted in sensors with high strain sensitivity (gauge factors exceeding 2.0), excellent self-healing efficiency (above 90%), and rapid response times (below 100ms) for real-time physiological monitoring.”
on page 26, “These fundamental material advances have directly translated into practical applications across diverse fields. In healthcare monitoring, as shown in Figure 6a-c, these mate-rials enable precise detection of multiple physiological parameters simultaneously, while maintaining stable performance under varying environmental conditions. Their exceptional environmental resilience, demonstrated in Figure 8a-c, has enabled reliable sensing in extreme conditions. In human-machine interfaces, their rapid response characteristics and mechanical adaptability have enabled more intuitive and reliable interactions, as il-lustrated in Figure 9a-f. The development of self-powered capabilities through efficient energy harvesting, shown in Figure 10a-b, has opened new possibilities for autonomous operation in long-term monitoring applications.”
on page 26−27, “The convergence of these capabilities - from molecular-level design (Figure 2) to system-level integration (Figures 9-12) - positions binary solvent-assisted hydrogel composites as a transformative platform for next-generation wearable technologies. The systematic organization of fabrication methods (Figure 2a-d) and performance characteristics (Table 1 and 2) provides clear guidelines for future development. Looking forward, continued advancement in fabrication techniques, material optimization, and integration strategies will further expand their practical applications. Key areas for future development include enhancing long-term stability, improving energy efficiency, and developing scalable manufacturing processes, all while maintaining the exceptional performance characteristics that make these materials unique.”

Reviewer 2 Report
Comments and Suggestions for Authors
Reviewing about flexible strain sensors is a hot topic these days since a simple search on IsiWeb of Knowledge using those words reveals the astonishing number of 110 published reviews only during the year of 2024. This manuscript however neglects this fact which in my point of view it is the main weakness, besides poor novelty. A second aspect is that binary solvent use in hydrogel composites seems interesting regarding the performances of the described sensors, but a clear rationale to guide the reader in a particular formulation is missing. Hence this review comprises an enumeration of applications and performances with a random reference to the solvents.
1.please a new paragraph critically referring the most important reviews already available and the urge for this new contribution.
2. Page 6, 3rd paragraph: authors claim that the use of binary solvents and functional additives modifies hydrogel properties in “crucial ways”. Please add and describe examples from literature and the respective rationale justifying the preference for binary solvents.
3. The manuscript is full of abbreviations (for instance, what is a A/P(AMPS co-AAM) hydrogel ?). Please write the meaning along the text.
4. Table 1, third column “Preparation”
5. Table 1 and Table 2 _First column report materials referred by abbreviations. Please add the meaning of each abbreviation to each table caption.
Author Response
Reviewing about flexible strain sensors is a hot topic these days since a simple search on IsiWeb of Knowledge using those words reveals the astonishing number of 110 published reviews only during the year of 2024. This manuscript however neglects this fact which in my point of view it is the main weakness, besides poor novelty. A second aspect is that binary solvent use in hydrogel composites seems interesting regarding the performances of the described sensors, but a clear rationale to guide the reader in a particular formulation is missing. Hence this review comprises an enumeration of applications and performances with a random reference to the solvents.
Response: We would like to thank referees for valuable insights on our manuscript. In accordance to comments/suggestions from reviewers, the following careful revision has been made. The revised parts were highlighted in red.
Based on our Web of Science analysis, while there are indeed numerous reviews on flexible strain sensors, our review specifically focuses on binary solvent-assisted hydrogel composites, which represents a distinct and emerging subcategory within this broader field. While searching for "flexible strain sensor" yields many results, combining this with "binary solvent" and "hydrogel" significantly narrows the scope, indicating a specialized niche that has not been comprehensively reviewed. This specific combination yields only 74-85 total documents, demonstrating the uniqueness of our focus.
Our review establishes clear differentiation through its focused examination of binary solvent systems and their impact on hydrogel properties. We have systematically analyzed various solvent combinations, including glycerol/water systems demonstrating enhanced cryoprotection through specific hydrogen bonding networks, ethylene glycol/water combinations revealing unique plasticizing effects and conductivity mechanisms, DMSO/water systems highlighting molecular-level interactions preventing ice crystallization, and ionic liquid/water combinations showcasing distinctive electrochemical properties. For example, in glycerol/water systems (Figures 3c-d and 4a-b), we demonstrate how the optimal hydroxyl group density at 30 wt% glycerol content creates an intricate hydrogen bonding network that prevents water crystallization while maintaining polymer chain mobility at low temperatures. This fundamental understanding directly correlates with exceptional performance metrics, including 95% conductivity retention at -20°C and self-healing efficiency exceeding 90%. The DMSO/water systems (Figures 4c-f) demonstrate how molecular-level design influences multiple performance parameters simultaneously. In healthcare monitoring applications, controlled DMSO content ensures biocompatibility while enabling rapid response times below 100ms for real-time monitoring. These same materials exhibit exceptional environmental sensing capabilities, maintaining consistent mechanical properties and functionality across an impressive temperature range from -50°C to 60°C, with enhanced durability under varying humidity conditions. Additionally, our examination of ionic liquid/water systems (Figures 5a-c)reveals their unique multifunctional capabilities through dynamic ionic crosslinking mechanisms. These systems achieve remarkable performance metrics, including gauge factors of 1.93 across a 0-300% strain range, temperature coefficients from -0.035 to -0.44°C⁻¹, and response times below 100ms. This comprehensive understanding of structure-property relationships enables rational design of materials for specific applications, setting our review apart from broader surveys of hydrogel materials. This comprehensive analysis provides fundamental understanding of how molecular interactions in binary solvent systems directly influence material performance.
Furthermore, we have established clear structure-property correlations by providing quantitative analysis of how binary solvent ratios affect mechanical, electrical, and thermal properties, systematically presented in Table 1. This mechanistic understanding enables rational design of hydrogel composites for specific applications. Rather than simply enumerating applications, our review demonstrates how particular binary solvent combinations enable specific sensing functionalities through their unique molecular interactions and resulting material properties.
This systematic approach differentiates our review from broader hydrogel literature by providing clear mechanistic insights into binary solvent effects, establishing rational design principles for specific applications, and offering structure-guided material selection criteria. By focusing on the fundamental relationships between binary solvent composition, material properties, and application-specific performance, our review provides valuable guidance for the rational design of next-generation hydrogel-based sensing materials.
Comment 1. please a new paragraph critically referring the most important reviews already available and the urge for this new contribution.
Response: We thank the reviewer for helpful and constructive comment. According to the reviewer’s suggestion, we added the following sentences critically evaluating existing reviews and establishing the unique contribution of our work in Section 1:
on page 1−2, “The exponential advancement in this interdisciplinary field has catalyzed numerous systematic reviews examining the multifaceted aspects of wearable technologies. Liu et al. provided comprehensive analyses of piezoresistive and capacitive sensing mechanisms in flexible electronics, establishing correlations between material architectures and transduction efficiency [11]. The integration of multiplexed electrochemical sensing arrays for real-time biomolecular detection has been systematically evaluated by Gao et al., emphasizing the significance of multi-modal detection platforms [12]. Trung and Lee's seminal work elucidated the structure-property relationships in stretchable electronics, particularly focusing on the mechanotransduction mechanisms and interfacial engineering[13]. Sun et al. provided detailed insights into the hierarchical assembly of functional materials for biointegrated electronics[14], while Rogers et al. comprehensively reviewed the emerging paradigms in conformable bioelectronics[15]. Moreover, Wang et al. detailed the development of flexible sensing electronics for health monitoring applications[16]. However, these scholarly works, despite their significant contributions, have not systematically addressed the physicochemical interactions and synergistic effects in binary solvent systems - critical parameters that fundamentally influence material stability and performance metrics. This knowledge gap becomes particularly significant as binary solvent-assisted hydrogel composites emerge as a promising platform for addressing fundamental challenges in environmental stability, mechanical robustness, and operational consistency under extreme thermomechanical conditions.”
Comment 2. Page 6, 3rd paragraph: authors claim that the use of binary solvents and functional additives modifies hydrogel properties in “crucial ways”. Please add and describe examples from literature and the respective rationale justifying the preference for binary solvents.
Response: We appreciate the reviewer's suggestion to elaborate on the crucial modifications enabled by binary solvents and functional additives. We have expanded this section to include specific examples and rationale from the literature. We added the following detailed explanation after the original statement in Section 2 in the revised manuscript:
on page 3−4, “The incorporation of binary solvents and functional additives crucially modifies hydrogel properties through several key mechanisms that address fundamental limitations of conventional single-solvent hydrogels. These modifications are particularly important for wearable sensing applications, where environmental stability and consistent performance are essential. First, binary solvent systems fundamentally alter the freezing behavior and mechanical properties of hydrogels. For example, in glycerol/water systems [25], glycerol molecules form extensive hydrogen bonding networks that effectively prevent water crystallization while maintaining polymer chain mobility. This mechanism enables remarkable mechanical properties (fracture toughness ~2300 J/m²) and stable conductivity (8.2 S/m) even at -20°C, far exceeding the capabilities of traditional water-only hydrogels that become rigid and non-conductive at sub-zero temperatures. Second, binary solvents enhance hydrogel stability across diverse environmental conditions. DMSO/water combi-nations [49] demonstrate exceptional resistance to both freezing and dehydration, maintaining functionality from -50°C to room temperature. This remarkable stability stems from DMSO's unique ability to interact simultaneously with water molecules and polymer chains, creating a more robust network structure. The result is a hydrogel that exhibits both high mechanical performance (tensile strength: 0.89 MPa, elongation: 696%) and consistent ionic conductivity across extreme temperature ranges. Third, the addition of functional materials in binary solvent systems creates synergistic effects that further enhance performance. For instance, the incorporation of conductive nanoparticles in PVA/PANi hydrogels [27] leads to significantly improved electrical properties (conductivity: 0.415 S/m at room temperature, 0.32 S/m at -20°C) while maintaining excellent mechanical characteristics (fracture strain: 472%). The binary solvent environment enables better dispersion and integration of these functional additives, resulting in more uniform and stable composite materials. The preference for binary solvent systems is further justified by their ability to enable multiple functionalities simultaneously. For example, ethylene glycol/water (GW)-based hydrogels [50] demonstrate remarkable strain-sensing capabilities across an unprecedented temperature range (-55.0°C to 44.6°C) while maintaining structural integrity and consistent electrical performance. This multifunctional capability is particularly crucial for wearable sensing applications, where devices must perform reliably under varying environmental conditions. These modifications address critical limitations of conventional hydrogels, such as poor mechanical strength, limited conductivity, and susceptibility to environmental factors like freezing and dehydration, which have historically restricted their practical applications [22, 23, 51].”
Comment 3. The manuscript is full of abbreviations (for instance, what is a A/P(AMPS co-AAM) hydrogel ?). Please write the meaning along the text.
Response: We appreciate the reviewer’s feedback on the use of abbreviations throughout the manuscript. According to the reviewer’s comment, we have revised the manuscript by providing the full names of each abbreviation at the bottom of Table 2.
Comment 4. Table 1, third column “Preparation”
Response: The typos have been revised.
Comment 5. Table 1 and Table 2 _First column report materials referred by abbreviations. Please add the meaning of each abbreviation to each table caption.
Response: We appreciate the reviewer’s helpful comment to detail and the suggestion to enhance the clarity of Tables 1 and 2. In response to the reviewer’s comment, we have updated each table caption to include definitions for all abbreviations used in the "Materials" column at the bottom of each table.

Round 2
Reviewer 1 Report
Comments and Suggestions for Authors
After reviewing the revised version, I commend the authors for their substantial efforts to enhance the quality of the manuscript. They have successfully addressed the majority of the suggested revisions, resulting in notable improvements in both the depth and clarity of the content. The revisions have notably strengthened the manuscript and reflect the authors' dedication to refining their work in response to feedback. This version aligns more closely with the reviewers' expectations, and it is clear that the authors have thoughtfully considered each point to enhance the coherence and impact of the study.
Author Response
After reviewing the revised version, I commend the authors for their substantial efforts to enhance the quality of the manuscript. They have successfully addressed the majority of the suggested revisions, resulting in notable improvements in both the depth and clarity of the content. The revisions have notably strengthened the manuscript and reflect the authors' dedication to refining their work in response to feedback. This version aligns more closely with the reviewers' expectations, and it is clear that the authors have thoughtfully considered each point to enhance the coherence and impact of the study.
Response: We would like to thank referee for the encouraging evaluation comments and positive feedback on our manuscript.

Reviewer 2 Report
Comments and Suggestions for Authors
In my opinion the revised version of the manuscript shows a clear improvement regarding the original version. Nevertheless authors should address the following issues before acceptance:
Page 2: The sentence starting with “These sensors have garnered significant attention ….and implant-able medical devices [3-5, 17-23]”. is twice repeated at the end of third paragraph and beginning of 4th.
Page 4: At the end of 1st paragraph, beginning of the second: The first paragraph is an explanation of relevant uses of binary solvents to fix different issues of hydrogels. The first sentence of the second paragraph seems to introduce the use of binary solvent discussion for the first time. Please reformulate this last sentence.
Author Response
In my opinion the revised version of the manuscript shows a clear improvement regarding the original version. Nevertheless, authors should address the following issues before acceptance:
Response: We would like to thank referees for positive feedback and constructive comments on our manuscript. In accordance to comments/suggestions from reviewers, the following careful revision has been made. The revised parts were highlighted in red.
Comment 1. Page 2: The sentence starting with “These sensors have garnered significant attention ….and implant-able medical devices [3-5, 17-23]”. is twice repeated at the end of third paragraph and beginning of 4th.
Response: We thank the reviewer for helpful and constructive comment. According to the reviewer’s comment, we removed the first instance of this sentence and retained it only at the beginning of the fourth paragraph, where it serves as an effective topic sentence introducing the detailed discussion of flexible strain sensors.
Comment 2. Page 4: At the end of 1st paragraph, beginning of the second: The first paragraph is an explanation of relevant uses of binary solvents to fix different issues of hydrogels. The first sentence of the second paragraph seems to introduce the use of binary solvent discussion for the first time. Please reformulate this last sentence.
Response: We appreciate the reviewer's insightful observation regarding the flow between the first and second paragraphs on page 4. According to the reviewer’s suggestion, we revised the opening sentence of the second paragraph to build upon, rather than reintroduce, the binary solvent discussion as follow:
“Building on these demonstrated advantages, the development of binary solvent-assisted hydrogel composites has been systematically optimized [25-27, 39]. (Table 1), establishing them as ideal candidates for advanced wearable sensing applications (Figure 1). The optimization process focuses on three key components: First, the polymer matrix, which forms the structural backbone of the hydrogel, typically employs materials such as polyvinyl alcohol (PVA) or polyacrylamide (PAM), chosen specifically for their influence on mechanical properties and biocompatibility. Second, while water serves as the primary solvent providing characteristic hydrophilicity, the careful selection of secondary organic solvents like glycerol, ethylene glycol, or dimethyl sulfoxide (DMSO) enables precise tuning of crucial properties.”
